# Mapping the sugar dependency for rational generation of a DNA-RNA hybrid-guided Cas9 endonuclease

Fernando Orden Rueda [1], Michal Bista[1], Matthew D. Newton[2], Anne U. Goeppert[1], M. Emanuela Cuomo[1], Euan Gordon[3], Felix Kröner[4], Jon A. Read[1], Jonathan D. Wrigley[1], David Rueda [2] & Benjamin J.M. Taylor [1]

The CRISPR–Cas9 RNA-guided endonuclease system allows precise and efficient modification of complex genomes and is continuously developed to enhance specificity, alter targeting and add new functional moieties. However, one area yet to be explored is the base chemistry of the associated RNA molecules. Here we show the design and optimisation of hybrid DNA–RNA CRISPR and tracr molecules based on structure-guided approaches. Through careful mapping of the ribose requirements of Cas9, we develop hybrid versions possessing minimal RNA residues, which are sufficient to direct specific nuclease activity in vitro and in vivo with reduced off-target activity. We identify critical regions within these molecules that require ribose nucleotides and show a direct correlation between binding affinity/stability and cellular activity. This is the first demonstration of a non-RNA-guided Cas9 endonuclease and first step towards eliminating the ribose dependency of Cas9 to develop a XNA-programmable endonuclease.

[1] Discovery Sciences, IMED Biotech Unit, AstraZeneca, Cambridge, UK. [2] Department of Medicine, Molecular Virology and MRC London Institute of Medical Sciences, Imperial College London, London W12 0NN, UK. [3] Discovery Sciences IMED Biotech Unit, AstraZeneca, Gothenburg, Sweden. [4] Dynamic Biosensors GmbH, Lochhamer Strasse 15, 82152 Martinsried, Germany. Fernando Orden Rueda and Michal Bista contributed equally to this work. Correspondence and requests for materials should be addressed to B.J.M.T. (email: Benjamin.Taylor@astrazeneca.com)

Genome engineering of mammalian cells has been rapidly advancing in recent years due to the adoption of the CRISPR–Cas9 RNA-guided DNA endonuclease system. Cas9 forms part of the bacterial and archaeal antiviral defence mechanisms, which degrade genomes of invading pathogens[1]. The most well-characterised version, the type II Cas9 protein derived from Streptococcus pyogenes (for simplicity, hereafter referred to as Cas9), relies on two associated RNA molecules to achieve activity and specificity: (1) a 42-nucleotide (nt) RNA molecule referred to as 'clustered regularly interspaced short palindromic repeat' RNA (also CRISPR RNA or crRNA) and (2) an 88-nt RNA molecule with constant sequence, known as trans-activating crRNA (tracrRNA). The crRNA has a 20-nt 'spacer' or 'guide' sequence at the 5′-end, which is variable and homologous to the intended DNA target sequence, and a 'repeat' sequence at the 3′-end that duplexes with the tracrRNA. These RNA molecules bind and span both the recognition and nuclease lobes of the 160 kDa Cas9 protein, the latter of which contains the HNH and RuvC nuclease domains[2]. Cas9 nuclease activity is a multistep process involving large domain movements and reorganisation[3–6]. Cas9 scans the genome searching for a 3 base-pair (bp) NGG sequence via the protospacer adjacent motif (PAM) recognition domain. Binding is transient unless both the crRNA and tracrRNA are complexed, at which point the DNA target is sequentially unwound and tested for complementarity with the crRNA. If near full complementarity is achieved, the HNH domain and recognition lobe undergo a conformational change to bring the nuclease sites into correct proximity for DNA cleavage (Supplementary Fig. 1 depicts the final complex). Following cleavage, Cas9 remains bound to the DNA, possibly releasing the PAM distal strand[7]. As Cas9 remains target-bound after cleavage, it has been classified as a single-turnover enzyme[8].

There has been interest in understanding the requirements and sequence specificity of the crRNA and tracrRNA molecules to help delineate the mechanism of Cas9 activity and allow improved exploitation and manipulation of this system. Both RNA molecules can be fused to create a fully functional single guide RNA (sgRNA)[9,10], and further sequence alterations and deletion studies have improved both transcription and stability as well as added new functional moieties[9–11]. Alterations to the nucleotides within these RNA molecules, such as 2′-O-methyl additions to the 2′-hydroxyl, have been found to reduce degradation and enhance nuclease activity in primary and immortalised cell lines[12]. These studies sought to further improve the sgRNA platform, yet there have been no reports to date that have addressed alternative nucleotide chemistry. The only chemical differences between RNA and DNA are a single hydroxyl group at the 2′ location on the sugar molecule and an extra methyl group on the thymine base, which substitutes for uracil. The hydroxyl group leads to altered properties, with RNA showing an increased preference for C3′-endo over C2′-endo sugar pucker, resulting in the more symmetric base-stacked A-form helix rather than the asymmetric B-form structure preferred by DNA[13,14]. In addition to the structure, differing backbone chemistries also alter the physical properties of nucleotide duplexes[13,14].

RNA is expensive to synthesise and the 2′-hydroxyl group in the ribose ring increases rates of hydrolysis and degradation, limiting its application and use. Improving these properties would not only yield cost benefits in the generation of CRISPR reagents and libraries, but may represent a crucial step in advancing the technology and leveraging its therapeutic potential. Further, the high stability of RNA:DNA duplexes led to the excess energy model for designing improved Cas9 systems: reducing the target-binding energy does not perturb on-target activity but significantly reduces off-target activity[15–17]. We therefore postulated that by manipulating the nucleotide backbone and fine tuning the duplex stability through modified nucleotide chemistry, it may be possible to significantly modulate many characteristics of the system, such as binding affinity, specificity, stability, in vitro and cellular activity, which would be crucial for development of efficacious CRISPR-based therapeutic entities.

To this aim, we here describe the first, to our knowledge, example of hybrid DNA–RNA CRISPR and tracr molecules that direct specific Cas9 nuclease activity in vitro and in vivo and possess unique biophysical properties, which could be exploited in future applications.

## Results

**A DNA CRISPR molecule cannot direct Cas9 nuclease activity.** To understand the molecular composition of the co-associated RNA molecules needed by Cas9, the requirement for RNA rather than DNA nucleotides within the CRISPR molecule was investigated. We generated a specific crRNA and an equivalently sequenced DNA molecule, a crDNA (CRISPR molecules will be

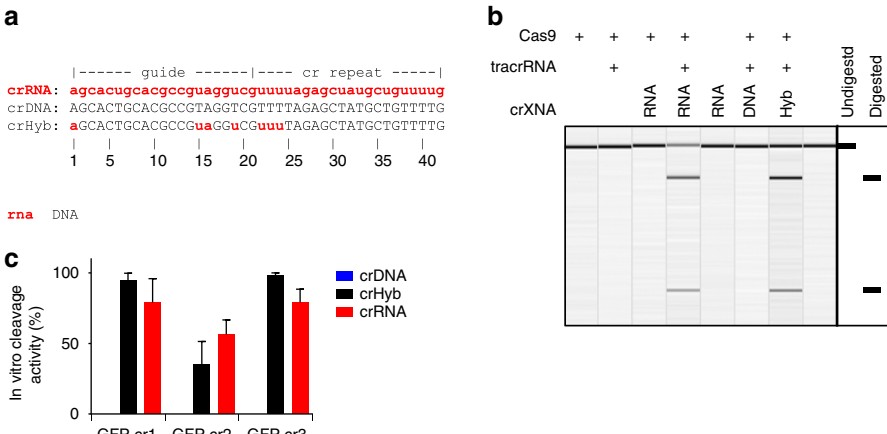

**Fig. 1** Cas9:tracrRNA complexed with RNA or hybrid crXNAs are nuclease competent. **a** Design of the crRNA, crDNA and crHyb sequence for GFP target site 1 (GFP cr1). DNA shown in uppercase black, RNA shown in lowercase red. **b** Cas9, tracrRNA and crXNA molecules were incubated with linear double-stranded DNA harbouring the GFP target sequence. Reaction products were analysed by agarose gel electrophoresis. The expected undigested and digested fragments are exemplified on the right. **c** Semiquantitative analysis of the cleavage of the linear DNA fragment by crDNA (blue)-, crHyb (black)- and crRNA (red)-guided Cas9:tracrRNA complexes with three independent guide sequences. Bars show mean + SD, n = 3–7

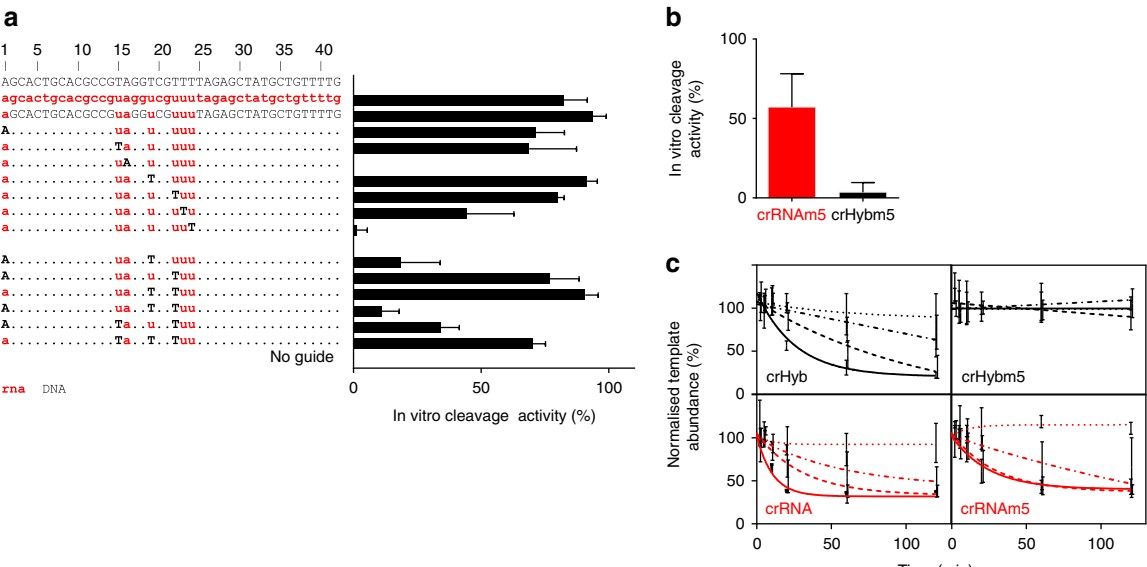

Fig. 2 Cas9:tracrRNA can function with a crXNA composed of 88% DNA and crHyb show enhanced specificity. **a** RNA nucleosides within the crHyb for GFP target 1 were sequentially converted to DNA and activity assessed by DNA fragmentation assay. Nucleotides coloured as in Fig. 1a. **b** A single base C > A mismatch was introduced into crRNA and crHyb molecules and activity assessed. All bars show mean + SD, n = 3–13. **c** Kinetics of template degradation driven by Cas9:tracrRNA complexed with crHyb (black) or crRNA (red), either with perfect complementary (left) or with a single base-pair mismatch (right). Graphs show normalised template abundance (%) over time, with Cas9 protein at 5 nM (solid line), 2.5 nM (long dash line), 1.25 nM (dot dash line) or 0.625 nM (dotted line). Error bars show SD of 3–4 independent measurements

collectively referred to as crXNA hereafter; Fig. 1a). These crXNA were designed to direct Cas9 to asymmetrically cleave a 790 bp double-stranded DNA (dsDNA) molecule, encoding enhanced green fluorescent protein (eGFP), allowing assessment of nuclease activity by DNA fragmentation. The assay was performed with excess crXNA and tracrRNA to ensure full saturation of Cas9 protein, and used a limiting DNA target concentration to allow for sensitive detection of cleavage activity. Cas9 protein alone was unable to cleave the target molecule and required both crRNA and tracrRNA for full activity, with specificity of cleavage shown by generation of fragments of the expected size; 533 and 258 bp (Fig. 1b). However, when a Cas9:tracrRNA:crDNA complex was used, no cleavage product was detected, indicating that ribose sugar moieties are essential for in vitro nuclease activity.

**A hybrid DNA–RNA crXNA can direct Cas9 cleavage.** The absence of nuclease activity when using a crDNA molecule could be due to loss of essential interactions with Cas9. Crystal structures of the Cas9:tracrRNA:crRNA complex with target DNA[18] reported 32 hydrogen bonds between Cas9 and crRNA (Supplementary Fig. 2); seven interactions rely on the ribose specific 2′ hydroxyl group (residues 1, 15, 16, 19, 22, 23 and 24 of crRNA), of which four are located in the guide sequence, where the nucleotides bind the target DNA, and three in the cr repeat region, which duplex the tracrRNA (Supplementary Figs. 2 and 3). To test whether these interactions were required for activity, a hybrid DNA–RNA crXNA molecule was generated, where 35 residues were coded as DNA and the 7 which make 2′ hydroxyl bonds with Cas9 protein coded as RNA (hereafter referred to as crHyb; Fig. 1a). The Cas9:tracrRNA:crHyb complex was nuclease competent in the DNA fragmentation assay (Fig. 1b). A crHyb molecule possessing only 7 RNA residues is therefore sufficient to direct specific Cas9 nuclease activity in vitro.

The replacement of uracil with thymine introduces extra methyl groups into the crXNA molecule. This methyl group should not impact base stacking with target DNA or tracrRNA. The position of these groups within the crXNA was manually

modelled within the Cas9 crystal structure[2] and we determined that no methyl group would come within 5 Å of protein. This suggests the presence of thymine bases would have no impact on the activity of the complex. To validate these predictions, we developed a quantitative PCR (qPCR) assay to report on template depletion by Cas9 cleavage using guide sequences and a 1 kb template sequence derived from the human AAVS1 locus. We generated a crHyb.deoxy molecule, where all thymidine nucleosides were replaced with deoxyuridine, to remove any additional methyl groups compared to an equivalent crRNA molecule. Assessing the ability of Cas9 complexes to cleave target DNA, we found template depletion over time was unchanged when Cas9: tracrRNA was complexed with crHyb or crHyb.deoxy, irrespective of Cas9 protein concentration (Supplementary Fig. 4). Thus, thymine within the crXNA does not appear to impact Cas9 cleavage activity.

**Cas9:tracrRNA binds stably to ribose-free crXNA.** To understand whether loss of activity was due to an inability of crDNA to complex with Cas9 protein, surface plasma resonance (SPR) was employed to measure binding kinetics of crXNA molecules to Cas9:tracrRNA complexes. Catalytically inactive Cas9 (dCas9; D10A and H840A) was immobilised to the SPR chip surface and associated with tracrRNA; dCas9 showed nanomolar affinity for tracrRNA with slow dissociation kinetics ($t_{1/2} \sim 0.5$ h; Supplementary Fig. 5a). Following assembly of this complex, binding kinetics of crXNA molecules were measured; crRNA, crHyb and crDNA showed comparable, high-affinity binding to the dCas9: tracrRNA complex in the subnanomolar range (Supplementary Fig. 5b, c) in line with previous measurements for sgRNA-binding affinities[19]. Furthermore, the specificity of crHyb binding to Cas9: tracrRNA was demonstrated by alteration of the cr repeat sequence to abrogate duplexing with tracrRNA, which rendered this crHyb (crHyb.duplex.mut) incapable of interaction (Supplementary Fig. 5d). Therefore, Cas9:tracrRNA complexes couple to crRNA, crDNA or crHybs with similar biophysical properties.

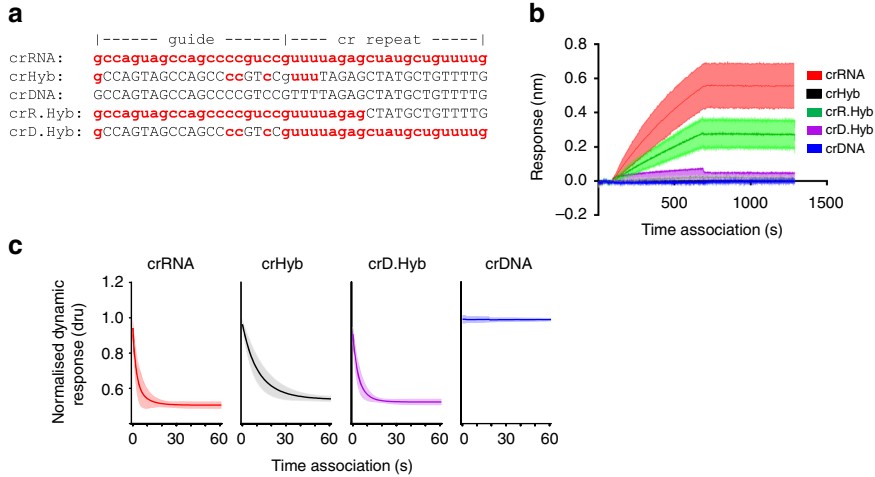

**Fig. 3** DNA in the crXNA molecule affects target-binding affinities. **a** Nucleoside composition of AAVS1 crXNA molecules; RNA shown in lowercase red, DNA in uppercase black. **b** BLI sensogram showing association kinetics to immobilised AAVS1 target DNA of Cas9:tracrRNA complexed with crRNA (red), crHyb (black), crDNA (blue), crR.Hyb (green) or crD.Hyb (purple). Shaded areas show SD of three independent measurements. **c** SwitchSENSE sensogram showing association phase of Cas9:tracrRNA:crXNA complexes binding to target DNA. Shaded areas show SD of four independent measurements

The finding that crDNA can bind to dCas9:tracrRNA complexes with broadly similar affinities to crRNA suggested that the correct holocomplex can form, but that it is not competent for either nuclease activity or target DNA recognition. Interestingly, we were unable to detect binding of a dsDNA target sequence when dCas9:tracrRNA:crXNA was immobilised on the SPR chip, possibly due to conformational restrictions imposed by immobilisation and covalent crosslinking.

**crHybs direct sequence-specific Cas9 nuclease activity**. Cas9 displays exquisite sequence-specific nuclease activity when directed by a crRNA. To determine whether such specificity was perturbed by crXNAs, three different regions of the same 790 bp DNA molecule were targeted. It was found that both crRNAs and crHybs showed robust in vitro nuclease activity at all target sites (Fig. 1c). Moreover, the cleavage products generated by crRNA and crHybs were identical, suggesting crHybs maintain the sequence specificity of Cas9 targeting (Supplementary Figs. 6 and 7). As expected, all three crDNA molecules generated no cleavage products.

**Identification of critical Cas9:crXNA contact points**. To determine whether all seven RNA nucleosides were required for Cas9 activity, each nucleoside in turn was sequentially converted to DNA and assessed for in vitro cleavage activity (Fig. 2a). crHybs containing either residues 1, 15, 19 or 22 as DNA nucleosides showed almost complete target DNA cleavage. Conversion of residue 23 to DNA led to a reduction of activity by 50%, whereas conversion of residues 16 or 24 to DNA led to an almost complete ablation of activity (Fig. 2a, top).

This analysis was extended by generating new crHybs with combinations of two or three of the five DNA nucleosides shown to have no adverse effect on activity. Synergies between which nucleosides could tolerate coding as DNA were detected; coding residues 1 and 22 or 19 and 22 as DNA had little effect on activity, whereas combining residues 1 and 19 seriously comprised activity with cleavage reduced below 20% (Fig. 2a, bottom). Combinations coding three residues as DNA were tested, but these were found to result in either moderate or severe reduction in activity. In summary, we have identified three key residues within the crHyb molecule (residues 16, 23 and 24), which cannot be coded as DNA. We have further described a fully active hybrid

design that contains only five RNA residues (residues 1, 15, 16, 23 and 24).

**crHybs show reduced off-target activity**. It has been demonstrated that reducing the helicase activity in the Cas9–target interaction increases the energy barrier needed to overcome mismatches between the sgRNA and target DNA, leading to increased specificity of the enzyme[16,20,21]. We questioned whether replacement of crRNA with the predominantly DNA coded crHyb would lead to similar increased specificity due to the reduced binding energy of a DNA–DNA duplex destabilising the complex[13]. We generated a crRNA and crHyb, which contained a C > A mutation at residue 5 (located in the non-seed region and which showed no sensitivities for encoding as RNA or DNA) and tested for activity on the unmodified DNA template. Whilst fully matched crRNA showed almost complete cleavage of the target DNA (Fig. 2a), the C > A mismatched crRNA showed a 30% reduction in activity (Fig. 2b). However, while the fully matched crHyb showed almost total cleavage of target DNA, when a single C > A mismatch was introduced the nuclease activity was almost completely ablated, with <4% activity detected (Fig. 2b). This suggests crHybs possess similar on-target activity but reduced off-target activity compared to crRNAs.

To determine whether the increased specificity was due to lower overall activity of crHybs, we performed time-course cleavage assays at multiple protein concentrations (Fig. 2c). For these experiments we used the qPCR template depletion assay described previously. We found clear dose-dependent and time-dependent cleavage activity with Cas9:tracrRNA:crRNA and Cas9:tracrRNA:crHyb complexes, with the latter showing overall slower template depletion kinetics. We found that a single base-pair mismatch in the crRNA resulted in only a slight perturbation of cleavage activity with a two- to fourfold reduction in the rate constant. However, the mismatched crHyb showed no detectable activity at any time point or concentration, demonstrating a much greater perturbation in catalytic activity.

**Cas9 target engagement is severely compromised with crHybs**. To determine if coding the crXNA molecule predominantly with DNA caused reduced binding affinity for target DNA, the kinetics of the complex interaction were measured. BioLayer Interferometry (BLI) and SwitchSENSE assays were developed to

monitor engagement of precomplexed dCas9:tracrRNA:crXNA with immobilised target DNA. For these assays, short target sequences derived from the human AAVS1 locus were used with crXNAs containing the corresponding 20-nt guide sequence (Fig. 3a). Both assays were able to validate binding of the dCas9:tracrRNA:crRNA complex to target DNA with high affinity and negligible dissociation (Fig. 3b, c, Supplementary Fig. 10 and Supplementary Table 2). However, an equivalent dCas9:tracrRNA:crDNA complex showed no interaction with target DNA (Fig. 3b, c), which corroborates the previously described absence of cleavage activity when using crDNAs.

Previous reports have shown that ionic blockade of Cas9 protein, by the negatively charged molecule heparin, can affect binding affinities[8]. In our BLI assay we similarly found polyanionic heparin reduced binding affinities and rates of association (Supplementary Fig. 11a). However, when heparin was absent, there was an increase in non-specific interaction between Cas9 (with or without associated RNA molecules) and specific or non-specific target DNA (Supplementary Fig. 11b). Addition of heparin completely abolishes binding to non-target sequences as well as non-specific interaction of apoCas9 with DNA.

Intriguingly, while both BLI and SwitchSENSE demonstrated efficient target recognition for Cas9:tracrRNA:crRNA complexes and no recognition for crDNA equivalents, the binding data for crHyb varied between the two methods. Using SwitchSENSE (Fig. 3c), we determined that crHybs had intermediate affinities, with on-rates slower than the crRNA. Whereas in the BLI assay (Fig. 3b), crHybs were significantly less able to promote dsDNA recognition (Fig. 3b, c and Supplementary Table 1).

To determine whether the reduced target-binding affinity was caused by a reduction of RNA in the guide or the cr repeat sequence, we generated two additional crXNA molecules; crR.Hyb and crD.Hyb (Fig. 3a). crR.Hyb coded only solvent accessible residues in the cr repeat region as DNA. This assumed that there would be no conformational restraints imposed by the protein on this region and it could therefore tolerate the altered backbone properties. crD.Hyb consists of a predominantly DNA guide region and fully RNA cr repeat region. Interestingly, we found that target-binding affinity was modulated by residues in the guide region. The BLI assay showed that crD.Hyb displayed weak target binding very similar to crHyb. Conversely, crR.Hyb showed an almost complete restoration of binding affinity (Fig. 3b). Using the orthogonal SwitchSENSE assay, despite crHyb performing better than in the BLI assay, crD.Hyb showed binding kinetics more similar, albeit still reduced, to crRNA. Thus, the ribose residues within the guide area, rather than the solvent exposed repeat region, appear most critical for target binding.

**Stable target engagement is compromised with crHyb complexes.** Having found that crXNA-binding affinities are reduced with increasing DNA content, we then sought to determine whether formation of the active complex is compromised. We used a previously developed single-molecule Förster resonance energy transfer (FRET) assay on our home-built total internal reflection fluorescence microscope to monitor dCas9 binding, conformational changes and dynamics[5,6,22]. Using biotin–streptavidin interactions, Cy3-labelled DNA containing the AAVS1 target site was surface-immobilised on a microscope slide (Fig. 4a and Supplementary Table 3; 5′ Bio-AAVS1-Cy3). We injected dCas9:tracrRNA:crXNA complexes with Cy5-label at the 5′-end of the crXNA (Fig. 4a and Supplementary Table 3). In this assay, binding of the dCas9 complex to the DNA yields a FRET signal between the donor (Cy3) and acceptor (Cy5), the magnitude of which is dependent on its conformational state (Fig. 4b and Supplementary Fig. 12).

With crRNA guide in the dCas9 complex, the majority of single-molecule trajectories (66%, n = 60/91) reach a stable high-FRET state (~ 0.95) which, in traces where the initial binding

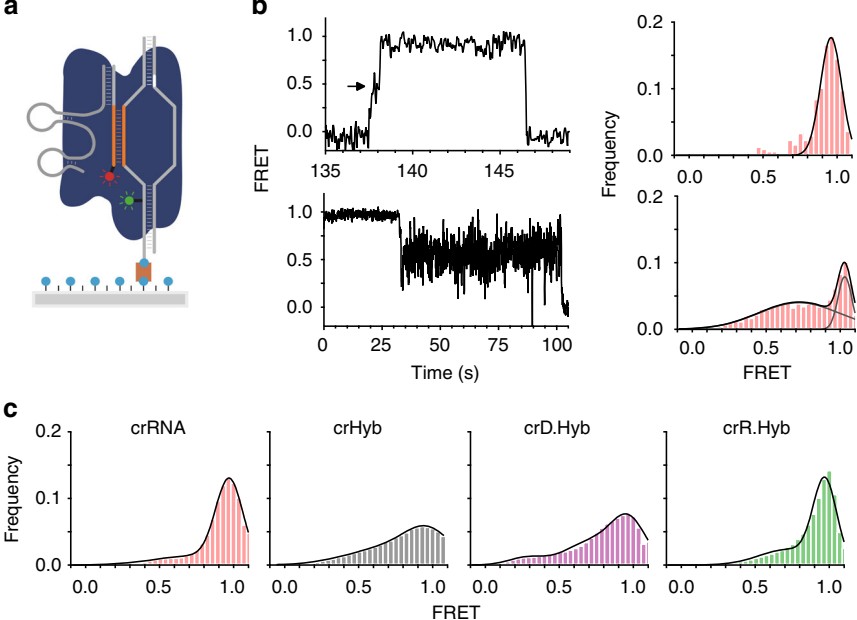

**Fig. 4** Conformational dynamics of target-bound dCas9:trRNA:crXNA complexes. **a** Single-molecule FRET experimental design. Surface-immobilised Cy3-labelled (green) DNA containing the AAVS1 target site (orange) and PAM site (white) binds Cy5-labelled (red) dCas9 complex (blue) to yield a FRET signal. **b** Representative single-molecule FRET trajectories show dCas9–crRNA complex binding in heteroduplex conformation as an increase to high FRET (top) through a transient intermediate (arrow). Irreversible conformational change into a dynamic intermediate FRET state (bottom). Corresponding time-binned FRET histograms (right), and Gaussian fits (black curves). Trajectories are 5 point averaged. **c** Time-binned FRET histograms for each guide: crRNA (red, n = 91); crHyb (grey, n = 57); crD.Hyb (magenta, n = 197), and crR.Hyb (green, n = 172)

 5

event is observed, transition through a transient intermediate FRET conformation (~ 0.5, arrow) with an average dwell time of $310 \pm 40$ ms ($n = 30$, Fig. 4b, top left). Based on previous studies, we assigned these trajectories to initial binding in an open conformation (~ 0.5 FRET, arrow) followed by zipping into a full heteroduplex conformation (~ 0.95 FRET)[5,6]. A single-molecule time-binned FRET histogram shows that these molecules predominantly occupy the high-FRET conformation, as expected (Fig. 4b, top right). Interestingly, a second population of trajectories (34%, $n = 31/91$) reach the high-FRET conformation but then transition irreversibly into a dynamic ensemble of states characterised by rapid FRET fluctuations from ~ 0.4 to ~ 0.9 FRET (Fig. 4b, bottom left). The corresponding time-binned single-molecule histogram shows a narrow peak centred on ~ 1.0 and a broad peak at ~ 0.70 (Fig. 4b, bottom right). This population may correspond to molecules that either unzip from the fully duplexed conformation and repeatedly attempt to return to it, or a protein conformational change that affects the local environment of the FRET pair. This population of molecules may not have been observed in previous studies, due to our work being at near-physiological [$Mg^{2+}$] (1 mM) rather than 10 mM in previous studies[5,6]. A time-binned FRET histogram of 91 trajectories with crRNA shows that in the presence of a fully RNA guide the complex predominantly exists in the fully heteroduplex conformation (Fig. 4c, red), consistent with the tight binding previously observed.

Increasing the DNA load in the dCas9 complex with a crHyb guide resulted in eightfold fewer overall binding events, consistent with the lower binding affinity observed by BLI and SwitchSENSE (Fig. 3b). Of the few binding events detected ($n = 57$), we observed a fourfold decrease of stable high-FRET trajectories (17.5%) in favour of traces demonstrating the dynamic broad mid-FRET state (70%) and a minor (12%) previously unobserved low-FRET (~ 0.2) population (Supplementary Fig. 12c). The corresponding time-binned FRET histogram shows a significant decrease in the high-FRET population and a broadening of the mid-FRET distribution (Fig. 4c, grey), indicating that high DNA load in crHyb hinders the formation of the active high-FRET conformation. These results are in agreement with the perturbed binding kinetics observed (Fig. 3). Taken together, the data with dCas9:tracrRNA:crHyb confirm the reduced binding affinity for target DNA and suggest that, in concordance with the detected in vitro cleavage activity, it can reach the active conformation required for cleavage but that this state has reduced stability, in agreement with the reduced cleavage activity in cells (Fig. 5).

To test whether the crHyb-binding defect arises from DNA in the guide region or the cr repeat region, we measured the binding dynamics of the Cas9:tracrRNA:crD.Hyb complex, (Fig. 3a). Interestingly, the data show a partial (3-fold) recovery of the number of binding observations compared to crHyb, but still (2.6-fold) less binding than crRNA, also in agreement with the BLI and SwitchSENSE measurements (Fig. 3b, c and Supplementary Table 1). However, only 18.5% of trajectories maintained the stable high-FRET state, while 53% display the dynamic broad mid-FRET state and the remaining 27% reach the 0.2 FRET state. The resulting time-binned FRET histogram (Fig. 4c, magenta, $n = 197$) shows only a minimal recovery of the high-FRET peak (compared to crHyb and crRNA), and a minor new peak at ~ 0.2 FRET. These data indicate that while crD.Hyb allows the Cas9 complex to efficiently bind target DNA, the active high FRET conformation remains unstable, which could explain its low activity in cells (Supplementary Table 1).

We similarly tested crR.Hyb (Fig. 3a). The dCas9:tracrRNA:crR.Hyb complex displays binding comparable to the full RNA guide (crRNA) complex, in agreement with the BLI and SwitchSENSE measurements (Fig. 3b, c and Supplementary Table 1). The fraction of trajectories reaching the stable high-FRET is 54% ($n = 93/172$) and a time-binned histogram shows an almost identical distribution to that of crRNA (Fig. 4c, green), both of which indicate that the crR.Hyb dCas9 complex adopts the stable high-FRET conformation with similar efficiency as the crRNA complex, in agreement with the observed cleavage activity in cells (Supplementary Table 1).

Overall, the single-molecule results suggest that DNA in the cr repeat region has little effect on the ability of the complex to transition from the 'open' to stably bound active state. In contrast, the presence of DNA in the guide region results first in an overall reduced binding affinity and second, when binding occurs, in a reduced stability of the high-FRET state in favour of the dynamic mid-FRET state. This suggests that the presence of high DNA load in the guide region destabilises this state resulting in rapid transitions between the two.

**Tight target engagement is required for cellular activity**. The reduced binding affinity and activity led to the question of how efficiently crHybs would work within the complex environment of mammalian cells. A mammalian cell reporter assay was established, using de novo mutations at the human AAVS1 locus as a readout of Cas9 nuclease activity. The AAVS1 locus is frequently used for genome modification and integration events; alterations at this site are not thought to affect cell viability[23]. Repair of the double-strand break caused by Cas9 leads to error-prone non-homologous end joining repair and base insertion or deletion (InDel) events. These events can be detected by PCR and Sanger sequencing of the target site followed by TIDE analysis to deconvolute the sequence traces and determine mutation frequency[24]. To introduce these modified crXNAs into HEK293 cells, electroporation of pre-formed Cas9:tracrRNA:crXNA complexes was undertaken. Five days after transfection, InDel generation at the AAVS1 target site was measured (Fig. 5).

Cells transfected with Cas9:tracrRNA:crRNA resulted in populations where over 60% of the AAVS1 loci contained InDels, indicating efficient targeting of Cas9 nuclease activity to this site. However, the equivalent Cas9:tracrRNA:crDNA complex caused no InDel formation, in agreement with our previous in vitro analysis. Cells transfected with Cas9:tracrRNA:crHybs also generated no InDels at the AAVS1 site, suggesting that the weak target-binding affinity is incompatible with targeting nuclease activity in a complex cellular environment.

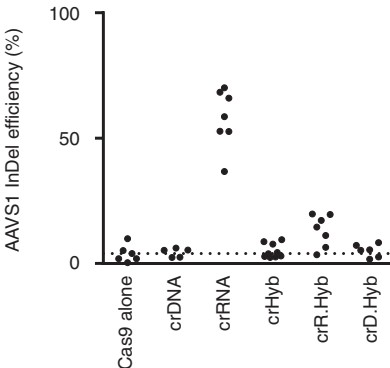

**Fig. 5** RNA-rich hybrid crXNA molecules can direct specific Cas9:tracrRNA nuclease activity in cells. Quantification of insertion and deletion (InDel) events at the AAVS1 locus 5 days after electroporation of Cas9:tracrHyb.v2:crXNA complexes. Dotted line show threshold determined from crDNA samples

**Table 1 Split tracr fragments sequences**

| Split tracr fragment | Sequence. RNA in lowercase and DNA in uppercase | DNA/ RNA | Design principal |
|---|---|---|---|
| tracr5′ RNA | caaaacagcauagcaaguuaaaauaaggcuaguccguuaucaacuugccuaguggcc | | |
| tracr5′ hyb v1 | CAAAACAGCATAGCAAguuAaAATAAGGCTAgTCCguuaTCaACTTGCCTAGTGGCC | 47/10 | Structure-guided design |
| tracr5′ hyb v2 | caaaacagcauagcAAguuAaAATAAGGCTAgTCCguuaTCaACTTGCCTAGTGGCC | 33/24 | 5′-anti-repeat as RNA |
| tracr5′ hyb v3 | CAAAACAGCATAGCaaguuaaaauaaggcuaguccguuaucaacuugccuaguggcc | 14/43 | Stem loop 1 & 2 as RNA |
| tracr5′ hyb v4 | CAAAACAGCATAGCaaguuaaaauaaggcuaguccguuaucaaCTTGCCTAGTGGCC | 28/29 | Stem loop 1 as RNA |
| tracr5′ hyb v5 | CAAAACAGCATAGCaaguuaaaATAAGGCTaguccguuaucaacuugccuaguggcc | 22/35 | ½ stem loop 1 and all stem loop 2 as RNA |
| tracr5′ hyb v6 | CAAAACAGCATAGCAAguuaaaauaaggcuaguccguuaucaaCTTGCCTAGTGGCC | 30/27 | Scanning modification of v4 |
| tracr5′ hyb v7 | CAAAACAGCATAGCaaguuaaaaTAAGGCuaguccguuaucaaCTTGCCTAGTGGCC | 34/23 | |
| tracr5′ hyb v8 | CAAAACAGCATAGCaaguuaaaauaaggcTAguccguuaucaaCTTGCCTAGTGGCC | 30/27 | |
| tracr5′ hyb v9 | CAAAACAGCATAGCaaguuaaaauaaggcuagTCCguuaucaaCTTGCCTAGTGGCC | 31/26 | |
| tracr5′ hyb v10 | CAAAACAGCATAGCaaguuaaaauaaggcuaguccguuaTCaaCTTGCCTAGTGGCC | 30/27 | |
| tracr3′ RNA | ggccacuaggcaaguggcaccgagucggugcuuuuu | | |
| tracr3′ hyb | GGCCACTAGGCAAgTgGCACCgAGTCGGTGcTTTTT | 36/4 | |
| tracr3′ DNA | GGCCACTAGGCAAGTGGCACCGAGTCGGTGCTTTTT | 40/0 | |
| tracrHyb.2 | GTTGGAACCATTCAAAACAGCATAGCaaguuaaaauaaggcuaguccguuaucaaCTTGAAAAAgTgGCACCgAGTCGGTGcTTTTT | 54/33 | |

To address this, cells were exposed to Cas9:tracrRNA:crR.Hyb and Cas9:tracrRNA:crD.Hyb. The cellular InDel efficiencies followed the same pattern as the binding kinetics data, with crD.Hyb showing no nuclease activity and crR.Hyb generating mutations as expected. However, InDel generation by crR.Hyb was fourfold less efficient than crRNA, with an average InDel frequency of 15.6%. In summary, in vitro functional hybrid crXNA molecules containing the minimal number of RNA residues are incompatible with cellular activity, whilst those with increased RNA content show activity, which is reduced compared to a fully RNA coded equivalent. This suggests that while tight target binding is essential for cellular nuclease activity, additional molecular characteristics are required for full efficiency.

**tracrHyb can support nuclease activity in vitro.** While the crRNA molecule is required for sequence-specific targeting of Cas9, the tracrRNA is the larger cofactor at twice the length of the crRNA molecule. The tracrRNA makes multiple hydroxyl contacts with Cas9 protein and duplexes with crRNA but is not thought to directly engage with target DNA[2]. Having demonstrated that crHybs were functional in vitro, we questioned whether the same would be true for the tracrRNA molecule, again with the potential for reduced synthesis costs and new properties. Following the same methodology as used to generate crHyb, a fully DNA tracr (tracrDNA) and a tracrHyb, which contained 10 RNA and 77 DNA residues were generated (collectively referred to as tracrXNA hereafter; Supplementary Fig. 13a). Using the same DNA fragmentation in vitro assay described previously, we found that neither of these molecules could support Cas9 nuclease activity when complexed with any crXNA molecule (Supplementary Fig. 13b).

To further dissect the backbone requirements for the tracrRNA and determine if any DNA residues could be tolerated, regions of the tracrHyb were reverted to RNA and tested for restoration of activity. To increase the experimental throughput, we first split the tracrRNA into two pieces; previous work has shown that truncations of the molecule which removed all residues from stem loop 1 onwards, were still compatible with in vitro nuclease activity[10]. The first fragment generated (tracr5′) contained the anti-repeat, stem loop 1 and stem loop 2 whilst the second

(tracr3′) covered stem loops 2 and 3 (Table 1). The length of the stem loop 2 duplex region was additionally increased to enhance stability of the duplexed split tracr (Fig. 6a). When complexed with Cas9 and crRNA or crHyb, the duplexed split-tracrRNA could direct equivalent cleavage activity to that of the single tracrRNA molecule (Fig. 6b). Interestingly, when only tracr5′ was used, full nuclease activity was detected in conjunction with crRNA but activity was reduced to over 10-fold for crHyb. Neither crRNA nor crHyb complexed with a hybrid version of tracr5′ generated cleavage products. Thus, while a crRNA can function with a minimal tracrRNA, the crHyb requires a full tracrRNA equivalent for activity.

Each tracrRNA fragment was replaced with a hybrid version to determine which could tolerate DNA nucleosides. No activity was detected when tracr5′ contained DNA nucleosides at non-hydroxyl interacting residues, whereas the combination of a hybrid tracr3′ and RNA coded tracr5′ gave full activity with both crRNA and crHyb (Fig. 6b). Therefore, a tracrXNA molecule partially encoded by DNA on the tracr3′ fragment can direct specific Cas9 nuclease activity in vitro.

Attention was then focused on the tracr5′ fragment and within the hybrid fragment framework we coded areas of stem loop 1, 2 and anti-repeat regions as RNA or DNA (tracr5′ Hyb v2-10; Fig. 6d and Table 1). Overall, we found that both the anti-repeat and stem loop 2 regions could tolerate DNA nucleosides (tracr5′ Hyb v4). Stem loop 1 was more sensitive to residue alteration (tracr5′ Hyb v5-10): we sequentially changed 2–4 bp of RNA to DNA across this region and could only identify two residues which could tolerate coding as DNA (tracr5′ Hyb v8). Interestingly, many of the tracrXNA molecules showed enhanced disruption of activity with crHyb compared to using crRNA (such as with tracr5′ Hyb v5, v7 and v9). Thus, stem loop 1 appears particularly sensitive to changes in tracrXNA structure, enhanced by DNA residues in the crXNA.

We inspected the Cas9:sgRNA crystal structure for potential steric clashes between the introduced thymine residues and the protein. All additional methyl groups were found over 5 Å from the protein, except the equivalent of nucleotide 90 where the group showed a clear clash (Supplementary Fig. 14a). However, it is possible that rotation of the backbone phosphate could induce a conformational change to relieve this clash, which correlates with

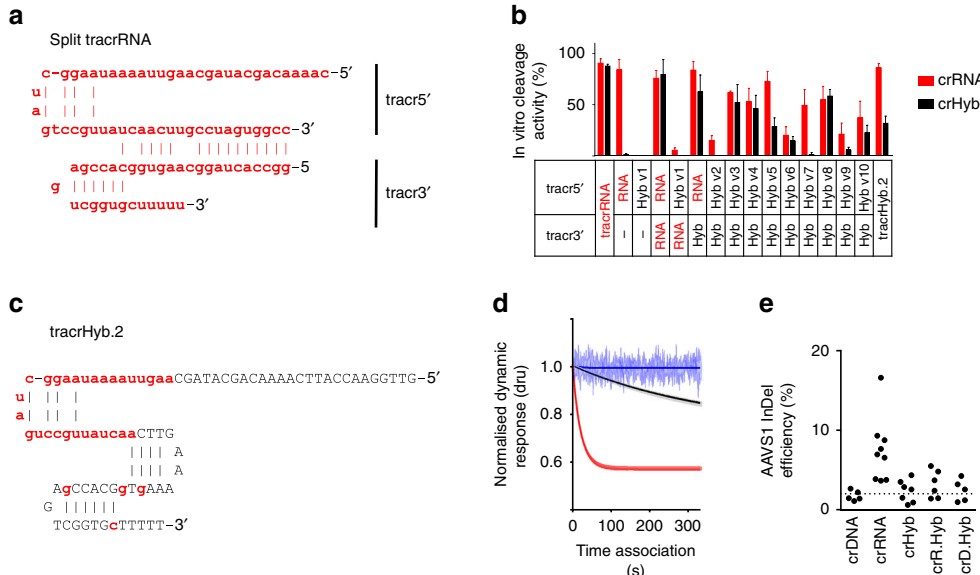

**Fig. 6** Biochemical and cellular characterisation of tracrHyb. **a** Design of the split-tracrRNA molecule. **b** Quantification of in vitro cleavage of target DNA by Cas9 complexes with crRNA (red) or crHyb (black) and combinations of single or split tracr molecules. **c** Design of the single tracr hybrid molecule (tracrHyb.2). **d** SwitchSENSE traces showing association phase of Cas9:tracrHyb.2:crXNA complexes binding to target DNA, coloured as in 3b. **e** Quantification of InDel events at the AAVS1 locus 5 days after electroporation of complexes composed of Cas9 and crRNA (red), crR.Hyb (green) or crDNA (black), and tracrRNA or tracrHyb.2. Dotted line shows threshold determined from crDNA samples and. For all charts, bars show mean + SD, $n > 3$

the observed cleavage efficiencies. In fact, this region could be coded completely as DNA, with little effect on cleavage efficiencies (Supplementary Fig. 14b, c), again, suggesting potential clashes in this region are tolerated.

In summary, we have successfully generated hybrid split tracr molecules, with the most efficient hybrid split tracrXNA molecules in biochemical assays coded by 50–70% DNA. Taking these design principals, the standard tracrRNA sequence was modified to incorporate DNA nucleosides identified as compatible from tracr5′ Hyb v4 and tracr3′ fragments. This generated a new tracrXNA hybrid (tracrHyb.2; Fig. 6c), which was 62% coded as DNA. In the DNA cleavage assay, a 1.4-fold reduction in activity was observed with crHyb when compared to the split tracrXNA equivalent; the now DNA encoded stem loop 2 is likely more constrained, which may destabilise the complex. However, when tested with crRNA, cleavage activity equivalent to a standard tracrRNA molecule was detected (Fig. 6b, significance tests shown in Supplementary Fig. 15). Thus, the tracrXNA can be modified to be a majority DNA encoded molecule and retain in vitro activity.

Biophysical characterisation of tracrHyb.2 using SPR revealed that binding to Cas9 is comparable to tracrRNA (Supplementary Fig. 16a and Supplementary Table 2). The SwitchSENSE assay revealed that target recognition of the Cas9:tracrHyb.2:crXNA complex had been significantly perturbed. The association kinetics, even using native crRNA, were significantly slower compared with tracrRNA (Figs. 6d and 3c). However, stable binding was observed for all complexes with slow or negligible dissociation as seen for tracrRNA (Supplementary Fig. 16b). These binding characteristics explain the differences seen in the cleavage assays, with the perturbed binding of Cas9:tracrHyb.v2:crHyb showing reduced in vitro cleavage efficiency.

**tracrHyb.2 is functional in eukaryotic cells**. Having generated a functional tracrHyb.2 molecule, activity was tested in the cellular assay. Following the electroporation assay as previously, we found that crRNA complexed with Cas9:tracrHyb.2 drove efficient InDel formation, although a third that of a fully RNA complex

(17% for Cas9:crRNA:tracrHyb.2 compared to 59% for Cas9:crRNA:tracrRNA; Figs. 5 and 6e). Cas9:tracrHyb.2:crR.Hyb was also able to generate InDels at low frequencies, whereas Cas9:tracrHyb.2 complexed with crDNA, crHyb or crD.Hyb was unable to efficiently generate InDels at the AAVS1 locus (Fig. 6e).

## Discussion

Cas9 is traditionally guided to its target by associated RNA molecules, which are inherently less chemically stable than DNA. Here we have described the design and optimisation of chemically synthesised hybrid DNA–RNA CRISPR and tracr molecules, which retain sequence-specific Cas9 endonuclease activity. Through rational design we have demonstrated that Cas9 can tolerate an almost complete change in nucleotide chemistry for the crXNA molecule, requiring just five RNA nucleosides for in vitro activity. These molecules can be used with Cas9 for in vitro site-specific nuclease purposes, provide a novel platform for cellular manipulation and allow development of more stable and cost-effective reagents for Cas9 applications. There have been novel in vitro uses of programmable endonucleases as restriction enzymes[25–27] and the DNA–RNA hybrid molecules described could certainly be adopted into such methods to take advantage of the improved stability and reduced synthesis costs. Further, crXNA and tracrXNA provide novel tools for interrogating Cas9 activity and structural requirements through careful manipulation of binding affinities and associated chemical moieties.

Cas9 tolerated the split tracr designs, with little impact on overall activity and provided a novel method to quickly assess designs for the final hybridised tracr molecule. We were successful in converting these split designs into a single tracr molecule that showed activities comparable to, or even higher than, the split versions; the increased activities could be due to improved folding of the single molecule. Interestingly, when complexing tracrHyb.2 with crHyb, activities were similar or lower than the split tracr counterparts. Such a decrease could be due to the additional constraints the deoxyribose backbone place on the conformation of stem loop 3. It would be interesting to

further probe this design by lengthening the loop to improve flexibility of the more rigid DNA molecule.

Through our combination of assays a clear correlation between target-binding affinity and cellular activity was found (results from all assays collated in Supplementary Table 1). However, while the rank order of activities of various Cas9-XNA combinations is maintained across the assays, the absolute activity of Cas9 was highly dependent on assay type and configuration. Biochemical in vitro cleavage assays and SwitchSENSE binding analysis were in good concordance, however both assays tended to overestimate the cellular activity of Cas9, which was best correlated with the association kinetics determined in the BLI assay and the number of binding events observed in the smFRET assay. There is a fundamental difference in DNA density used in those assays; both biochemical and SwitchSENSE methods use relatively low concentrations of target DNA and excess Cas9 complexes. Conversely, in both BLI and cellular assays, local DNA content greatly exceeds the number of Cas9 complexes, albeit predominantly on-target in the former and off-target in the latter. Due to this high DNA concentration, the BLI, but not the SwitchSENSE, assay required heparin to reduce non-specific interactions between Cas9 complexes and non-target DNA. We hypothesise that high DNA content can amplify the effect of non-specific unproductive binding of Cas9 outside its target sites. Such transient non-specific binding will compete and interfere with the search for regions of complementarity. In agreement with this notion, a recent computational study has suggested that the Cas9 complex undergoes many rounds of target binding and dissociation before nuclease competent binding occurs[28]. Thus, inherently frequent and incompetent binding events coupled with overall reduced affinities of crHyb molecules are therefore incompatible with efficient searching and target recognition of the ~6400 Mb human genome.

There are several possibilities for reduced target association rates of Cas9 loaded with crHyb molecules. First, the direct duplexing energies of DNA–RNA are higher than the DNA–DNA equivalents[13,14]. Inclusion of DNA in the guide region will affect the binding energies and thermodynamic stability of the final complex, which is fully supported by the biophysical data presented here. Second, Cas9 has evolved to function with a predominantly A-form DNA–RNA duplex within the catalytic pocket. While DNA–DNA duplexes can form both A or B form, the latter is preferred. It is possible that within the catalytic pocket, the target-guide duplex is presented in a suboptimal conformation, reducing binding stability. Third, recent single-molecule studies have demonstrated that mismatches in the guide sequence can stall Cas9 in an 'open' configuration part way through the target binding induced structural rearrangement, with the implication that this is due to reducing stability and binding energy in the duplex[5,6,21]. Our data support the notion of multiple steps of structural reconfiguration required for efficient target engagement. Whilst our measurement of tracrRNA and crXNA-binding affinities are not impacted by Cas9 protein immobilisation (Supplementary Fig. 16), we did find crosslinked Cas9 showed no interaction with target DNA, likely due to conformational constraints. The single-molecule FRET data highlight that weakened binding affinities and stabilities lead to a dynamic interaction with target DNA, likely though an inability to stably engage in the fully duplexed conformation. One avenue for improving activity with crXNA molecules would be through the use of alternative backbone chemistries that may be more tolerated by Cas9, or through evolution of Cas9 to accommodate these molecules more efficiently by improving the conformational changes of the complexes. Joung and colleagues have established a bacterial assay,[20] which could be adapted by electroporating synthetically generated crXNAs into bacterial cells inducibly

expressing Cas9 variants libraries. This would allow rapid identification of Cas9 mutants that function more effectively with crXNA in vivo.

Additional factors may also account for the reduced cellular activity of tracrHyb and crHyb. In vitro, a crHyb molecule should demonstrate improved stability over a RNA counterpart. However, the converse could be true in a cellular environment where the RNaseH family are ubiquitously expressed and will degrade any DNA:RNA duplex or DNA–RNA hybrid strand[29]. Thus, one open question is how protected these molecules are when complexed with Cas9 protein. Protection of crRNA ends by phosphate backbone modifications have been shown to improve editing efficiencies, suggesting that degradation by cellular enzymes can reduce the half-life of the complex within the cell. Such modification could be employed with crHybs as a potential area for further improving editing efficiencies.

We present here the first, to our knowledge, demonstration of Cas9 activity directed by a predominately RNA-free guide molecule. Further improvements to this system will be enhanced by full structural and conformational characterisation of Cas9:tracrXNA:crXNA complexes. Such characterisations will aid and direct novel variations within tracrXNA, crXNA or Cas9 protein to further tolerate altered backbone chemistries and improve binding properties and catalysis rates. We envisage these insights will open new avenues for further reductions in Cas9's ribose dependency and the generation of fully RNA-free endonucleases with novel properties, such as enhanced specificity and stability, which could be desirable when taking programmable guided nucleases into therapeutic settings.

## Methods

**Biochemical cleavage assay.** DNA templates were generated by PCR amplification using Phusion Flash High-Fidelity polymerase (Thermo Fisher) and the following primers eGFP_fw: 5′-GCT AGC CTC GAG AAT TCT GCA G-3′; eGFP_rv 5′-GCG GCC GCT TTA CTT GTA CA-3′; AAVS1_fw 5′-GCT TGC CAA GGA CTC AAA CC-3′ and AAVS1_rv 5′-GTC TTC TTC CTC CAA CCC GG-3′. The cleavage reaction of these templates contained 20 nM Cas9 (New England Biolabs), 60 nM tracrXNA (Dharmacon and Eurogentec), 60 nM crXNA molecules (Integrated DNA Technologies (IDT) and Eurogentec), 1× Cas9 reaction buffer (New England Biolabs) and 3 nM target DNA. The reaction was incubated at 37 °C for 1 h and terminated by 10 min incubation at 80 °C followed by desalting via dialysis across a cellulose membrane with a pore size of 0.025 μm (Merck-Millipore). Fragments were subsequently quantified by automated QIAxcel agarose gel electrophoresis (Qiagen). Cleavage activity was calculated by dividing the normalised area of the cleavage products by the normalised area of all fragments. Exemplar raw gels for Figs. 1 and 2b are shown in Supplementary Figs. 7 and 9, respectively.

For qPCR detection of template depletion, Cas9 cleavage assays were conducted using 5, 2.5, 1.25 or 0.625 nM Cas9 (New England Biolabs), 15 nM tracrXNA (Dharmacon and Eurogentec), 15 nM crXNA molecules (IDT and Eurogentec), 1× Cas9 reaction buffer (New England Biolabs), 0.75 nM eGFP and AAVS1 target DNA. Reactions were stopped after 0, 2, 5, 10, 20, 60 or 120 min by addition of equal volume of 100 mM EDTA, 0.1 ng ml$^{-1}$ Proteinase K solution, followed by heating to 65 °C for 5 min. Reactions were diluted 20-fold with water, and then used directly for qPCR using Luna qPCR Probe Master Mix (New England Biolabs) following the recommended protocol with the following set of primers and probes; GFP_fw; 5′-AAG TCG TGC TGC TTC ATG TG-3′, GFP_rv; 5′-GAG GAG CTG TTC ACC GGG-3′, GFP_probe; 5′-6FAM-ATG AAC TTC AGG GTC AGC TTG C-3′, AAVS1_fw; 5′-CCT GGT GAA CAC CTA GGA CG-3′, AAVS1_rv; 5′-ACC CTC TCC CAG AAC CTG AG-3′, AAVS1_probe; 5′-HEX-CTCCGTGCGT CAGTTTTACC-3′. The reactions were run on an ABI Prism 7900HT cycler and Cts determined. All samples were measured in triplicate. The derived concentration of the AAVS1 template was normalised to the input control GFP template using ΔΔCt method. Time courses were normalised to the starting concentration from the zero time point. qPCR assay efficiency is shown in Supplementary Fig. 8.

All crXNA sequences are given in the figures except GFP_crRNAm5; 5′-rArGrC rArArU rGrCrA rCrGrC rCrGrU rArGrG rUrCrG rUrUrU rUrArG rArGrC rUrArU rGrCrU rGrUrU rUrUrG-3′, GFP_crHybm5; 5′-rAGC AAT GCA CGC CGrU rAGG rUCG rUrUrU TAG AGC TAT GCT GTT TTG-3′, AAVS1_crRNAm5; 5′-rGrCrC rArArU rArGrC rCrArG rCrCrC rCrGrU rCrCrG rUrUrU rUrArG rArGrC rUrArU rGrCrU rGrUrU rUrUrG-3′ and AAVS1_crHybm5; 5′-rGCC AAT AGC CAG CCrC rCGT rCCG rUrUrU TAG AGC TAT GCT GTT TTG-3′.

**Cas9 protein production**. A synthetic gene coding for Cas9 or catalytically dead Cas9 (dCas9; D10A and H840A), with an N-terminal 6×HN tag (consisting of 6 histidine and asparagine pairs) and a C-terminal nucleoplasmin nuclear localisation sequence was synthesised and subcloned into pET24a via *Nde*I and *Xho*I restriction sites to generate pET24a-Cas9. *Escherichia coli* λBL21DE3* transformants of pET24a-Cas9 were selected on LB plates containing 100 μg ml$^{-1}$ kanamycin. Large-scale cultures were performed by inoculating 750 ml of TB media at 37 °C, with the cultures grown at 37 °C until OD$_{600}$ = ~ 0.6. The culture temperature was then lowered to 20 °C and protein production induced by the addition of 100 μM isopropyl β-D-1-thiogalactopyranoside. Incubation was continued overnight. Cas9 was purified from cell lysate via immobilised metal affinity chromatography. Briefly, cells were lysed using an Avestin Emulsiflex C5 in a buffer consisting of 20 mM Tris Cl (pH 7.5), 300 mM NaCl, 10% glycerol, 1 mM Tris(2-carboxyethyl)phosphine hydrochloride (TCEP) and 10 mM imidazole, and the lysate clarified by centrifugation at 20,000 × *g* for 20 min. The clarified lysate was loaded onto a 5 ml HisTrap column (GE Healthcare) and, after washing with the same buffer, eluted with a gradient to 300 mM imidazole. The Cas9-containing fractions were, after dilution, loaded onto a 5 ml Heparin HiTrap column (GE Healthcare) and eluted with a linear gradient of 0.1–1 M NaCl in a buffer containing 20 mM Tris (pH 7.5), and 10% glycerol. The protein was polished by size-exclusion chromatography, on a Superdex 26/60 column (GE Healthcare) in a buffer containing 20 mM HEPES (pH 7.5), 150 mM KCl, 10% glycerol and 1 mM TCEP. Fractions containing Cas9 were pooled and concentrated to 10–20 mg ml$^{-1}$ and aliquots flash frozen in liquid nitrogen.

**AAVS1 cell assay**. Synthetic crXNA (IDT or Eurogentec) and tracrXNA molecules (Dharmacon and Eurogentec) were mixed at equal molarities to give a 960 ng μl$^{-1}$ solution in duplex buffer (IDT), heated for 5 min at 100 °C and cooled to room temperature. A volume of 1 μl of the crXNA:tracrXNA mix was added to 1 μg of Cas9 protein (in-house purified) in a total volume of 5 μl of Neon Electroporation Buffer R (Thermo Fisher). The Cas9:tracrXNA:crXNA complexes were allowed to form for 10 min at room temperature in low-protein-binding tubes. HEK293 cells (ATCC) were harvested, washed with phosphate-buffered saline (PBS) and 3.5 × 10$^5$ cells were resuspended in 5 μl of Buffer R supplemented with 0.2 μg of Cas9 Electroporation Enhancer (IDT)[30]. The cells and Cas9:tracrXNA:crXNA complex solutions were mixed and electroporated using a 10 μl Neon tip, protocol 24 on the Neon Transfection System (Thermo Fisher). After electroporation, cells were seeded into a well of a 24-well plate and incubated for 5 days at 37 °C, 5% CO$_2$. Cells were then washed in PBS and lysed with DirectPCR buffer supplemented with 0.1 mg ml$^{-1}$ Proteinase K (Viagen Biotech) following the manufacturer's instructions. The AAVS1 target region was amplified by PCR using Phusion Flash High-Fidelity polymerase (Thermo Fisher) with the following primers: AAVS1_genome_fw 5′-GCT TGC CAA GGA CTC AAA CC-3′ and AAVS1_genome_rv 5′-GTC TTC TTC CTC CAA CCC GG-3′. PCR products were purified (illustra GFX PCR DNA and Gel Band Purification Kits; GE Healthcare Life Sciences) and Sanger sequenced. Total InDel efficiency was then calculated using TIDE analysis (https://tide.nki.nl/)[24] comparing samples with sequence traces from untransfected cells.

**Biophysical assays**. All biophysical assays were performed with in-house-purified Cas9 and dCas9 as follows.

*Surface plasmon resonance*: SPR measurements were carried out at 25 °C using a Biacore T200 instrument (GE Healthcare Life Sciences). dCas9 was immobilised via standard amine coupling on a CM5 chip using immobilisation buffer (50 mM HEPES (pH 7.0), 0.005% Tween-20). The typical immobilisation level observed was ~10,000 RU. Measurement buffer consisted of PBS supplemented with 0.005% Tween-20. Data were analysed with the BIAevaluation software (GE Healthcare Life Sciences). Oligonucleotides tested were as above and crHyb.duplex.mut 5′-rAGC ACT GCA CGC CGrU rAGG rUCT rArArA ATT ACG ATA CGA CAA AAC-3′.

For experiments with immobilised tracrHyb, the following oligonucleotide was used 5′-Biotin-TEG-GTT GGA ACC ATT CAA AAC AGC ATA GCrA rArGrU rUrArA rArArU rArArG rGrCrU rArGrU rCrCrG rUrUrA rUrCrA rACT TGA AAA ArGT rGGC ACC rGAG TCG GTG rCTT TTT-3′.

*Biolayer interferometry*: BLI measurements were performed at 25 °C using an Octet Red96 instrument (ForteBio). The measurement buffer consisted of PBS supplemented with 0.005% Tween-20, 1 mM MgCl$_2$ and 50 μg ml$^{-1}$ heparin. SA tips were used to immobilise ca. 0.1 unit (nm) of 5′-biotinylated duplex DNA oligo containing a sequence complementary to crXNA molecules (AAVS1-on-target 5′-biotin-CAC CCT CGT GAC CAC CCT GGA CGG GGC TGG CTA CTG GCT CAG CCG CTA CC-3′ and non-specific target 5′-biotin-CAC CCT CGT GAC CAC CCT GGA CGG GGC TGG CTA CTG GCT CAG CCG CTA CC-3′). Subsequently, tips were typically dipped in the measurement buffer for 180 s, transferred to precomplexed dCas9:tracrXNA:crXNA (formed at 100 nM:200 nM:200 nM) for a 30 min association step and finally moved to the measurement buffer for a 30 min dissociation step.

*SwitchSENSE*: SwitchSENSE experiments were performed on a DRX2 analyser using CST-48-2-G1R1-S chips (both Dynamic Biosensors GmbH, Martinsried, Germany)[31]. The chip featured two DNA anchors of different sequence (anchor A, 5′-TAG TGC TGT AGA AGA ATA GAC TCG CTC GTG TTG ACA AGA

ACT GAT-Green 3′ and anchor B, 5′-TCG TCT GTC TCA CTG ATG TAT GAT TAG TAA GAA CGC TCG CAC GCT GAT-Red 3′), which were optimised for minimal dCas9 interaction (PAM-free) as well as minimal cross-hybridisation. Anchors A and B lie in close proximity to each other (average distance 30 nm) on the chip's gold electrodes and are labelled at their 3′-end with a green fluorescent dye and a red fluorescent dye, respectively. dCas9 samples were prepared in T100K5 buffer (20 mM Trizma Buffer, 100 mM KCl, 5 mM MgCl$_2$ and 0.01% Tween-20, pH 7.4) and contained precomplexed dCas9:tracrXNA:crXNA (formed at 20 nM:30 nM:30 nM) and incubated at 37 °C for 20 min.

The kinetic measurements of each dCas9:tracrXNA:crXNA combination binding to DNA was performed using a standard kinetics assay from the switchBUILD software from Dynamic Biosensors GmbH. Before each dCas9:tracrXNA:crXNA injection, anchors A and B were stripped of their complementary strand with regeneration solution (pH 13, <1 s contact). Subsequently, anchor A and B were functionalised through hybridisation with a dsDNA overhang elongated by their respective complementary anchor sequence. The overhang on anchor A contained a reference sequence (Cas9Ref, 5′-AAA AAA AAA AAA AAA AAA AAA AAA AAG GAA-complementary anchor A-3′) while the overhang on anchor B contained the target sequence (Cas9 target, 5′-AAA AAG CCA GTA GCC AGC CCC GTC CAG GAA-complementary anchor B-3′). Each dCas9:tracrXNA:crXNA complex was injected at a concentration of 20 nM in T100K5 buffer with a constant flow of 50 μl min$^{-1}$. The association of dCas9:tracrXNA:crXNA with each of the strands was observed using the dynamic response of anchor A and anchor B in real time. Dissociations were performed for 10 min under a constant flow of 500 μl min$^{-1}$ T100K5 buffer.

Analysis was performed with the switchANALYSIS software from Dynamic Biosensors GmbH and Graphpad Prism 6.01. The association rate constants ($k_{on}$ and $k_{off}$) and the respective error values were derived from a single exponential fit model. All oligonucleotides were obtained from Biomers, Germany.

*Single-molecule FRET assay*: tracrRNA was synthesised by T7 RNAP transcription, and crXNA guides were synthesised, HPLC-purified and 5′-Cy5-labelled by IDT (sequences in Supplementary Table 3). Target DNA with 5′-biotin and internal amino linker modified T was synthesised, HPLC-purified by IDT and subsequently labelled with Cy3 NHS ester[32] and HPLC-purified in-house. All oligonucleotides were further purified by denaturing polyacrylamide gel electrophoresis, as previously described[22].

Flow chambers were prepared as previously described[33,34]. Briefly, quartz slides and coverslips were passivated with polyethylene glycol (5% biotinylated) and flow chambers constructed using double-sided sticky tape and sealed with epoxy. Pre-annealed dsDNA (final concentration 12.5 pM) was immobilised via biotin–streptavidin interactions. dCas9 complex was formed with 1 μM dCas9, 2 μM tracrRNA and 1 μM labelled crXNA at room temperature for 5 min before dilution to a final concentration of 10 nM with imaging buffer (50 mM Tris-HCl (pH 8), 100 mM NaCl, 1 mM MgCl$_2$, 0.2 mg ml$^{-1}$ bovine serum albumin) and oxygen scavenger system (5 mM 3,4-dihydroxybenzoate (PCA) and 100 nM protocatechuate-3,4-dioxygenase (PCD)) in saturated Trolox (~ 3 mM). The flow chambers were imaged on a home-built, prism-based total internal reflection microscope with a 532 nm excitation laser (~ 2 mW), and images acquired on an EM-CCD camera (Andor) with a 30 ms exposure time. FRET efficiencies were calculated from integrated donor ($I_D$) and acceptor ($I_A$) intensities as FRET = $I_A$/($I_D + I_A$)[33,34].

Images and data were analysed by custom IDL, MATLAB and R scripts, available upon request. Processed traces used for the transition analysis are given in Supplementary File 1.

**Data availability**. All the data generated in this study are available in the manuscript and accompanying documents or from the authors upon reasonable request.

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

## Acknowledgements

Thanks to the AZ Discovery Sciences Cellular Reagents and Transgenics teams for critical discussions. Additional thanks to AZ colleagues Claire McWhirter, David Fisher, Ander Gunnarsson, Mikko Hölttä, Marcello Maresca, Tim Kaminski, Martin Packer and Amir Taheri-Ghahfarokhi for helpful and insightful discussions. The Rueda lab is funded by a core grant of MRC London Institute of Medical Sciences (RCUK MC-A658-5TY10) and a start-up grant from Imperial College London. MDN is funded by a BBSRC studentship.

## Author contributions

F.O.R., M.B., M.D.N., F.K., A.U.G., E.G. and B.J.M.T. conducted experiments. M.B., M.E.C., J.A.R., J.D.W., D.R. and B.J.M.T. conceived and supervised the study. B.J.M.T., M.B. and D.R. wrote the manuscript. All authors reviewed and approved the manuscript.

## Additional information

**Competing interests:** The authors declare no competing financial interests.

