## [Peer Review File · Nature Communications]

Reviewers' comments:

Reviewer #1 (Remarks to the Author):

Review of Mapping Cas9's sugar dependency: rational generation of a DNA-RNA hybrid guided Cas9 endonuclease

In this manuscript, the authors attempt to use in vitro and in-vivo techniques to demonstrate the dependency of structure on the performance of CRISPR/Cas9. The methodology involved modifying the structure of DNA:RNA and tracr molecules based on a 'guided' approach. The authors used SPR, BLI and SwitchSENSE to determine binding affinity and thereby the activity for the dependency of Cas9 sugar in the molecular interaction process. While this is a very well written paper and the molecular biology seems to have been carried out properly, this reviewer questions the validity of the claims made in the paper. In the discussion, much of what was presumably measured / determined is called into question by the authors. To my mind, this observation puts the premise of the paper in question. In other words, it seems like they were unable to truly map the sugar dependency and make a reasonable correlation with structure. The major flaw is the "discontinuity between the absolute activity of Cas9 and the assay type / configuration. They state that there is a concentration dependence (which is not surprising given that true K_d values require the receptor concentration be much lower than the K_d). The need to use heparin in one of the assays seems somewhat problematic and the inability to control for non-specific binding due to DNA amplification should be a consideration for taking these observations as less than optimum. The authors are applauded for their recognition of the importance of structural changes in these biochemical transformations. What they have not fully recognized is the significant impact of immobilization of a target on the outcome of a binding assay. Recent work published by Koch and co-workers, and others have shown that SPR sensograms can be very misleading when significant binding-induced structural changes are produced. This is undoubtedly the case for BLI as well, which also demands surface immobilization, thereby changing the target molecule structure and constraining motion of the receptor during binding.

In summary, for this well written paper to rise the level of Nature Communication these limitations should be addressed with additional confirmatory experiments by definitively, 1) showing the K_d values have not been perturbed by the measurement technique; 2) confirming that structural reconfiguration is required for Cas9 during target binding, and 3) determining how the 'open configuration' impacts stability of the duplex.

- Figure 4 in the supplemental is a bit confusing to this reviewer, particularly given the potential problems with unraveling structural responses vs. binding (see above).

Reviewer #2 (Remarks to the Author):

In this work, the authors determine whether RNA nucleotides can be replaced with DNA nucleotides in the Cas9 gRNA sequence. They replace nucleotides both in the crRNA and tracrRNA. A large number of positions in both molecules could be swapped for DNA, while maintaining robust cleavage activity in vitro and even some cleavage inside cells. This work was certainly of interest and unique and the data was convincing. The writing and figures were also well-organized and clear. However, I think the accessibility would be greatly enhanced by the addition of structural figures to put the explanations of the system and engineering processes into a visual context. I have a few additional concerns and suggestions to improve the work, but nothing that significantly dampens my overall enthusiasm.

Major comments

- I think the story could be enhanced by more justification in the introduction about what this tool could be used for and why make it. Stability and cheapness are valid reasons, but it might help to

have some specific reasons why stability is important (gRNAs work very well as they are for most purposes). On a similar note, I think it would help to include a discussion of possible next steps in the discussion section. Would the next step be to engineer the Cas9 protein with computational design or directed evolution to accept this DNA molecule? If so, an appropriate method of directed evolution could be identified and cited (probably would want something inside cells, where it would be reasonable to use RNA/DNA hybrid gRNAs). If this is a direction that authors think is interesting, it's possible they could even include a final figure in the results section that identifies potential protein residues to target in the engineering.

- In light of the data regarding compromised target engagement for crHybs, I'm a bit concerned about the section stating crHybs are more sequence-specific. I think it would be safe to say that they are at least equally sequence-specific. The evaluation that they are more specific is based on a single time-point and enzyme concentration, rather than on actual enzymatic parameters. It is clear from the data following that the crHybs bind much more poorly to their target sites. However, the fact that the crHyb is clearly not as good at generating cleavage as crRNA is masked when the assay is conducted with the one Cas9 concentration and reaction time. If one were to compare say 1 minute cleavage reactions or cleavage with 3-fold lower Cas9 concentration, it is almost certain that the crHyb without the mismatch would perform substantially worse than the crRNA without a mismatch. Thus, the ratio of cleavage mismatched/matched cleavage may actually be the same for the crRNA and crHyb during many conditions.

- In relation to above comment: in the Methods section it says "the reaction was incubated at 37°C for stated times", but I can't seem to find these times for each of the experiments. Perhaps I missed them, but I checked within the main text, the figures, and figure legends and it would be helpful if they are more obvious. Are all reactions shown completed with the same incubation time?

- Using a more stringent reaction condition can also be used to determine which of the bases in the crHyb are causing problems for in vivo activity, by doing these tests with reverting single nucleotides. I don't think this experiment is necessary for publication, but it could help the authors with future efforts.

- While the use of these newer biophysical assays is interesting (even just as an assessment of the utility of these tools), since cleaving ability is the parameter of more interest why did the authors not do standard single-turnover enzyme kinetics to both get and binding (K_m) and turnover (k_{cat})? It is straightforward to do a time-course with a few enzyme concentrations. Based on the biophysical data presented, it appears likely that K_m and formation of the initial ground-state binding complex is hindered. However, confirming that turnover is also not changed would be important for future engineering, as it could influence the type of directed evolution used (would want something based on cleavage, not binding only).

- This work should include some figures (either main text or supplemental) that show the crystal structures and hydrogen bonds that were considered during the structure-guided engineering. Page 3 description of the hydrogen bonds that were considered during the engineering process should be accompanied by a figure, probably of those hydrogen bonds. Again on page 6 (line 230) where the RNA nucleotides were chosen for the tracrRNA, it would be helpful to have a figure to explain to the reader how the engineering choices were made. Both a picture of the relevant hydrogen bonds and then also one that shows the stem loops (page 6, line 239-241) would be useful.

- If the authors believe that one of the reasons certain bases cannot be substituted is the formation of interactions with the hydroxyl group, perhaps it would be worth testing some of these positions with LNA nucleotides? If an LNA nucleotide maintains activity, they could get much closer to their goal of a very stable molecule.

- I noticed that the Hyb v8 tracr pair actually has higher activity than the final tracrHyb.2. I wonder if actually mutating those two nucleotides to DNA is helpful or if it's a difference between the fused and unfused form of RNA. Perhaps the authors could comment on this or do a test.

- does the tracr3 half actually need to contain any RNA bases? I'm aware that some of those bases make contacts hydroxyl contacts with the protein, but did the authors ever try to fully remove all RNA from this portion of the tracr? It seems possible that this part of the tracr doesn't matter so much, especially given it can be truncated.

Minor comments

- To make this work more accessible to a reader that is not intimately familiar with the structure of Cas9 and the relationship between the tracrRNA and crRNA, I would suggest adding a supplementary structure figure. If I read this paper pretending I don't know how these molecules interact, I find that the explanations are insufficient to really visualize the system and experiments. For example, when tracrRNA is first introduced it is not clear to an unfamiliar reader that the tracrRNA weaves through the Cas9 protein.

- Page 5, line 204: tracrRNA should be tracrRNA

- Page 5, lines 146-156: There is confusion in the writing here about whether residue 15 can be changed to DNA or not. At line 146 the text indicates that it can be changed, but then it is not tested in any of the combinations listed and is called a key residue on line 156. Looking at the figure, it does appear to have more of an effect than 1, 19, and 22, but the writing needs to be clarified in the main text for consistency.

- Page 5, line 188 and Figure 3: It seems like the crR.Hyb contains RNA nucleotides in more than the RNA guide (target site recognition) region, as they also extend into the following RNA hairpin. Was a crR.Hyb tested that contained just the direct target site binding residues (and any others that are required, such as 23 and 24)? If not, what was the reasoning for extending the RNA residues past the target contact region? I don't think it's a necessary experiment, I just wonder about the reasoning or if there are additional data that weren't included in the manuscript.

- Page 7, line 279: I think this sentence needs a comma between "molecule" and "activity", or needs to be rewritten.

- Page 7, line 281: I would cite Figure 4 at the end of the sentence to remind readers where to find this data, and/or include numbers for comparison between the data in the two figures.

Reviewer #3 (Remarks to the Author):

In this manuscript Rueda and coworkers investigated the importance of the ribonucleotides in the CRISPR molecule for Cas9 binding and activity. They produced several DNA-RNA hybrids of crRNA and tracrRNA containing deoxyribonucleotides instead of ribonucleotides at different positions of the molecules and tested their effect on Cas9 binding, DNA target recognition and Cas9 nuclease activity in vitro and in vivo. Based on these data, they identified critical regions of the Cas9 system that require ribonucleotides. Such work could be used for developing artificial CRISPR/Cas9 systems for making programmable endonuclease that could be used for therapeutic purpose. In that context, this work can be very valuable. However, a number of major points need to be addressed before this work is published.

Major points:

The authors performed several U to T substitutions and systematically concluded that the absence of 2'OH was responsible for the observed effects on Cas9 binding/activity. However, this substitution also introduce a methyl at position 5 of the base. The possible effect of this change should be evaluated using the structure of the Cas9:tracrRNA:crRNA complex bound to DNA and discussed.

It is important to present raw data of cleavage activity assays (at least in supplemental data) in order to evaluate the quality of the results (e.g. Fig. 2B).

Dissociation curves of switchSENSE measurements are not shown (Fig. 3C and 5D). These results are important to judge about the quality of the data and visualize the absence of dissociation reported in the text.

Given the binding variations observed in Fig. 3 between BLI and switchSENSE measurements, it is important to repeat the experiments at different concentrations of Cas9:tracrRNA:crRNA to obtain standard deviations.

In Fig. 5D the final plateau corresponding to the complete saturation of DNA molecules by Cas9:tracrRNA:crRNA should be visible to properly evaluate the fit of the data and the correct determination of the k_{on} .

Minor point:

Figure 3C is not cited in the text.

Reviewers' comments:

Reviewer #1 (Remarks to the Author):

Review of Mapping Cas9's sugar dependency: rational generation of a DNA-RNA hybrid guided Cas9 endonuclease. In this manuscript, the authors attempt to use in vitro and in-vivo techniques to demonstrate the dependency of structure on the performance of CRISPR/Cas9. The methodology involved modifying the structure of DNA:RNA and tracr molecules based on a 'guided' approach. The authors used SPR, BLI and SwitchSENSE to determine binding affinity and thereby the activity for the dependency of Cas9 sugar in the molecular interaction process. While this a very well written paper and the molecular biology seems to have been carried out properly, this reviewer questions the validity of the claims made in the paper.

In the discussion, much of what was presumably measured / determined is called into question by the authors. To my mind, this observation puts the premise of the paper in question. In other words, it seems like they were unable to truly map the sugar dependency and make a reasonable correlation with structure. The major flaw is the "discontinuity between the absolute activity of Cas9 and the assay type / configuration. They state that there is a concentration dependence (which is not surprising given that true Kd value require the receptor concentration be much lower than the Kd). The need to use heparin in one of the assays seems somewhat problematic and the inability to control for non-specific binding due to DNA amplification should be a consideration for taking these observations as less than optimum.

- We thank the reviewers for their careful reading of the manuscript and interesting discussion points. Whilst the biophysical assays do indeed show differences in the measured Kds, there is a consistency in the rank order between assays (crRNA > crR.Hyb > crD.Hyb > CrHyb). Further, we propose that the assays actually provide interesting insights into Cas9 biology; with its high local DNA concentration, we suggest the BLI assays is more represented of a mammalian nucleus, with results correlating with the cellular assays. Conversely, the SwitchSENSE assays with low the Cas9 mediated DNA hydrolysis would require of the *in vitro* biochemical assay, with the binding results showing a similar correlation to the cleavage data with respect to crRNA and crHyb. One major issue that does confound these assays is the complex nature of the Cas9 catalytic cycle, with the multistep binding and cleavage events. Measuring the kinetics of each of these steps has yet to be demonstrated by anyone in the field and is beyond the scope of this manuscript. To support our conclusions, we have now included an additional assay based on single-molecule binding assays. We believe that we have presented enough data to demonstrate the development of functional hybrid crXNA and tracrXNA molecules and provide convincing biochemical, biophysical and cellular data which link their activity to the structural requirements of hydroxyl groups on the backbone, the stability and affinity of binding and the *in vitro* and *cellular* activities
- Regarding the use of heparin in the assays, this is a standard component of many Cas9 assays, and again we feel that the use of this reagent highlights important biological differences between the systems, that is when local DNA concentration is very high, there is a great deal of non-specific binding which requires a blocking agent. This high local DNA concentration would represent a nuclear environment and the addition of heparin would phenocopy the effect of the complex milieu of the nucleus where many molecules will block non-specific interactions of an exogenously introduced Cas9 molecule. We found that in vitro binding data generated in presence

of heparin correlates better with results of cleavage assays in cells, and there is also a good correlation between results of *in vitro* biochemical and biophysical binding assays executed in absence of heparin (Supplementary Table 1).

- Together, we hope these additional data and discussion addresses this reviewer's concerns.

The authors are applauded for their recognition of the importance of structural changes in these biochemical transformations. What they have not fully recognized is the significant impact of immobilization of a target on the outcome of a binding assay. Recent work published by Koch and co-workers, and others have shown that SPR sensograms can be very misleading when significant binding-induced structural changes are produced. This is undoubtedly the case for BLI as well, which also demands surface immobilization, thereby changing the target molecule structure and constraining motion of the receptor during binding.

In summary, for this well written paper to rise the level of Nature Communication these limitations should be addressed with additional confirmatory experiments by definitively,

1) showing the K_d values have not been perturbed by the measurement technique;

- To address this comment, we have performed additional SPR experiments using immobilised Cas9 or tracrHyb molecule immobilised by a conjugated biotin. This mitigates the potential for conformational restriction using immobilised Cas9. We find the affinities and binding kinetics of tracr molecules as SPR ligand or analyte are very similar, indicating that the assay setup does not significantly influence the results.
- Constraining dynamics of covalently immobilised receptors is valid point and our data showed that while immobilised Cas9 was able to bind tracrRNA and CRISPR, it was unable to recognise the target DNA. The latest step is known to involve gross structural rearrangement and covalent cross-linking is very likely to interfere with that process.
- We have included the additional data as Supplementary Figure 17 and the following text in the discussion:

“Our data supports the notion of a multiple steps of structural reconfiguration required for efficient target engagement; whilst our measurement of tracrRNA and crXNA binding affinities are not impacted by Cas9 protein immobilisation (Supplementary Fig. 16), we did find cross-linked Cas9 showed no interaction with target DNA, likely due to conformational constraints.”

- While we have been unable to formally demonstrate crXNA binding to Cas9:tracrRNA is unperturbed by assay conditions, we have demonstrated that in these series of assay systems, there is no binding of Cas9:tracrRNA complexes to target DNA. Therefore, the binding kinetics we do measure are a consequence of the full complex. Coupling this with the new single-molecule data which also show a significantly perturbed stability of target bound DNA, we feel the presented data is now sufficient to conclude that it is not Cas9:tracrRNA:crHyb complex formation that is deficient, but the ability of this complex to correctly engage with target DNA.

2) confirming that structural reconfiguration is required for Cas9 during target binding, and

- This point is intimately linked to question three, as such please see response below

3) determining how the ‘open configuration’ impacts stability of the duplex.

- We have performed single molecule FRET studies as outlined in the recent Lim et al., 2016 paper to indicate whether crHybs result in accumulation of stalled ‘open’

complexes similar to mismatched crRNAs. These results are presented in Figure 4 and discussed in the subsection 'Stable target engagement is comprised with crHyb complexes'. Our new data shows that crHyb forms an unstable complex when bound to target DNA, with a high rate of transition between a FRET states. This contrasts to crRNA which rapidly forms a stable high FRET conformation. We feel this additional data addresses both points 2 and 3 and highlights that there are multiple steps of structural reconfiguration during target binding. The biophysics data support the notion that both initial binding and the final conformational changes are affected.

- *Figure 4 in the supplemental is a bit confusing to this reviewer, particularly given the potential problems with unraveling structural responses vs. binding (see above).*

- We apologies for lack of clarity on the explanation of this figure. Supplementary Fig. 5 shows the SPR assays measuring tracrRNA binding (arrow 1 on panel A) followed by crXNA binding and dissociation (arrows 2 and 3 respectively on panel A). Panel D shows the specificity of this experiment as when the cr-repeat region of crHyb was mutated, no binding was observed, indicating that correct interaction between crHyb and tracrRNA was required. We have improved this figure to aid clarity.
- Regarding structure and binding, with the control data shown above showing the lack of impact of immobilisation on Cas9:tracrRNA interaction, we feel this assay allows estimation of the differences in affinity of crXNA for Cas9:tracrRNA complexes. We do see a slightly lower binding affinity of crHyb for the Cas9:tracrRNA complex than crRNA. However, this difference is not great enough to explain the differential activities seen across the range of binding and activity assays. Complemented with the new data shown in Figure 2c demonstrating improved cleavage specificity of crHybs and the single molecule data presented in Figure 4, we feel that the overall conclusion of this data is that differential activity of crHyb molecules are due to effects on the stability of target DNA binding rather than Cas9:tracrRNA:crXNA association.

Reviewer #2 (Remarks to the Author):

In this work, the authors determine whether RNA nucleotides can be replaced with DNA nucleotides in the Cas9 gRNA sequence. They replace nucleotides both in the crRNA and tracrRNA. A large number of positions in both molecules could be swapped for DNA, while maintaining robust cleavage activity in vitro and even some cleavage inside cells. This work was certainly of interest and unique and the data was convincing. The writing and figures were also well-organized and clear. However, I think the accessibility would be greatly enhanced by the addition of structural figures to put the explanations of the system and engineering processes into a visual context. I have a few additional concerns and suggestions to improve the work, but nothing that significantly dampens my overall enthusiasm.

- We thank the reviewer for their comments. We have sought to address all points and feel these amendments have strengthen the overall work.

Major comments

- I think the story could be enhanced by more justification in the introduction about what this tool could be used for and why make it. Stability and cheapness are valid reasons, but it might help to have some specific reasons why stability is important (gRNAs work very well as they are for most purposes).

- We have expanded this section and provided more rationale for the study. We have included the following:

“RNA is expensive to synthesise and the 2' hydroxyl group in the ribose ring increases hydrolysis and the rate of degradation, limiting its application and use; Improving these properties would not only yield cost benefits in the generation of CRISPR reagents and libraries, but may represent a crucial step in advancing the technology itself and leveraging its therapeutic potential. Further, the high stability of RNA:DNA duplexes has led to the excess energy model for designing improved Cas9 systems, whereby reducing the target binding energy does not perturb on-target activity but significantly reduces off-target activity¹⁵⁻¹⁷. We therefore postulated that through manipulation of the nucleotide backbone and fine tuning the duplex stability through modified nucleotide chemistry, it may be possible to significantly modulate many characteristics of the system, such as binding affinity, specificity, stability, *in vitro* and *cellular* activity, which would be crucial properties for development of efficacious CRISPR-based therapeutic entities.”

On a similar note, I think it would help to include a discussion of possible next steps in the discussion section. Would the next step be to engineer the Cas9 protein with computational design or directed evolution to accept this DNA molecule? If so, an appropriate method of directed evolution could be identified and cited (probably would want something inside cells, where it would be reasonable to use RNA/DNA hybrid gRNAs). If this is a direction that authors think is interesting, it's possible they could even include a final figure in the results section that identifies potential protein residues to target in the engineering.

- This is indeed a direction we have considered. We have now included the following text in the discussion:

“One avenue for improving activity with crXNA molecules would be through the use of alternative backbone chemistries that maybe more tolerated by Cas9, or through evolution of Cas9 to accommodate these molecules more efficiency and improve conformational changes of the complexes. Joung and colleagues have established a bacterial assay²⁰ which could be adapted to allow evolution and rapid screening to identify Cas9 variants which function more efficiency with crXNA *in vivo*.”

In light of the data regarding compromised target engagement for crHybs, I'm a bit concerned about the section stating crHybs are more sequence-specific. I think it would be safe to say that they are at least equally sequence-specific. The evaluation that they are more specific is based on a single time-point and enzyme concentration, rather than on actual enzymatic parameters. It is clear from the data following that the crHybs bind much more poorly to their target sites. However, the fact that the crHyb is clearly not as good at generating cleavage as crRNA is masked when the assay is conducted with the one Cas9 concentration and reaction time. If one were to compare say 1 minute cleavage reactions or cleavage with 3-fold lower Cas9 concentration, it is almost certain that the crHyb without the mismatch would perform substantially worse than the crRNA without a mismatch. Thus, the ratio of cleavage mismatched/matched cleavage may actually be the same for the crRNA and crHyb during many conditions.

- We have performed additional experiments as suggested with an orthogonal assay and set of reagents, shown in Figure 2c. We have found that while the rate constant is reduced with crHybs compared to crRNA, the further reduction that occurs when a mismatch is introduced is far greater with crHybs than crRNA. As the reviewer suggests, this ratio of mismatched/matched was very informative and we feel that this new data supports the conclusion that crHybs have increased specificity. We have included the following text:

“To determine whether the increased specificity was due to lower overall activity of crHybs, we performed time-course cleavage assays at multiple protein concentrations (Figure 2c). For these experiments, we used the qPCR template depletion assay discussed previously. We found clear dose dependent and time dependent cleavage activity with correctly matched Cas9:tracrRNA:crRNA and Cas9:tracrRNA:crHyb complexes, with the latter showing overall slower template depletion kinetics. We found that a single base pair mismatch in the crRNA resulted in only a slight perturbation of cleavage activity with a two to four-fold reduction in the rate constant compared to the fully complementary version. However, the mismatched crHyb showed no activity at any time point or concentration, demonstrating a much greater perturbation in catalytic activity due to a single base mismatch.”

- Importantly, other groups have theoretically and practically demonstrated that reducing the activity of cas9 by slowing down target recognition is one of strategies of increasing the fidelity of the enzyme (see <https://doi.org/10.1016/j.cels.2016.12.010>).

In relation to above comment: in the Methods section it says “the reaction was incubated at 37°C for stated times”, but I can’t seem to find these times for each of the experiments. Perhaps I missed them, but I checked within the main text, the figures, and figure legends and it would be helpful if they are more obvious. Are all reactions shown completed with the same incubation time?

- This was a mistake and has been amended.

Using a more stringent reaction condition can also be used to determine which of the bases in the crHyb are causing problems for in vivo activity, by doing these tests with reverting single nucleotides. I don’t think this experiment is necessary for publication, but it could help the authors with future efforts.

- The authors thank the reviewer for this helpful comment, but agree that it is beyond the scope of this current manuscript and is something we will certainly consider for future work.

While the use of these newer biophysical assays is interesting (even just as an assessment of the utility of these tools), since cleaving ability is the parameter of more interest why did the authors not do standard single-turnover enzyme kinetics to both get and binding (K_m) and turnover (k_{cat})? It is straightforward to do a time-course with a few enzyme concentrations. Based on the biophysical data presented, it appears likely that K_m and formation of the initial ground-state binding complex is hindered. However, confirming that turnover is also not changed would be important for future engineering, as it could influence the type of directed evolution used (would want something based on cleavage, not binding only).

- This is an excellent comment and indeed determining whether initial binding or turnover is affected by crHybs would be extremely valuable for future efforts.

- As Cas9 is a single turnover enzyme and does not obey Michaelis-Menten kinetics (Doudna et al Nature 2014), we were unable to determine the individual constants for k_{cat} and K_M .
- We have performed additional experiments at multiple time point and multiple protein concentrations as suggested. Using pre-steady state kinetic analysis, we were able to fit the single turnover data to $[S] = [S]_0 e^{-kt}$ where k is the apparent k_{cat}/K_M (specificity constant) for catalysis of RNA or crHyb Cas9 mediated cleavage. (Segel Enzyme Kinetics p41-43).
- Cas9 complexed with RNA was found to have a ~3.5 fold higher apparent k_{cat}/K_M than the crHyb:Cas9 complex. This is consistent with the observation that Cas9:tracrRNA has a higher affinity for RNA than crHyb however, separating the initial binding and turnover kinetics is complicated by the multiple intermediate reconfiguration steps which are required for catalysis.
- We currently do not have a clear understanding of which of these intermediate steps are affected by crHybs and as such, determining precisely which activity/step is affected is complex. Further kinetic characterisation to determine the rate constants for every step of the Cas9 mediated DNA hydrolysis would require many additional experiments and we feel such characterisation is beyond the scope of this paper.

This work should include some figures (either main text or supplemental) that show the crystal structures and hydrogen bonds that were considered during the structure-guided engineering. Page 3 description of the hydrogen bonds that were considered during the engineering process should be accompanied by a figure, probably of those hydrogen bonds. Again on page 6 (line 230) where the RNA nucleotides were chosen for the tracrRNA, it would be helpful to have a figure to explain to the reader how the engineering choices were made. Both a picture of the relevant hydrogen bonds and then also one that shows the stem loops (page 6, line 239-241) would be useful.

- We have added the suggested diagrams to Supplementary Fig.2. We have used a modified version of a Figure 5 from Nishimasu et al., Cell 2014. We would appreciate guidance on whether this is acceptable or presents a copyright issue.

If the authors believe that one of the reasons certain bases cannot be substituted is the formation of interactions with the hydroxyl group, perhaps it would be worth testing some of these positions with LNA nucleotides? If an LNA nucleotide maintains activity, they could get much closer to their goal of a very stable molecule.

- This is a very interesting comment. We have tried adding LNA residues to internal positions within the crHybs but found them to result in non-functional complexes. However, these modifications were not direct replacements for the RNA residues. Whilst this is an interesting comment, we feel that a full exploration of LNA and indeed other unnatural residues, with their altered structural and binding properties will be worthy of a separate manuscript and beyond the scope of this current work.

I noticed that the Hyb v8 tracr pair actually has higher activity than the final tracrHyb.2. I wonder if actually mutating those two nucleotides to DNA is helpful or if it's a difference between the fused and unfused form of RNA. Perhaps the authors could comment on this or do a test.

- crRNA showed significantly improved cleavage efficiencies when complexed with tracrHyb.v2 than Hyb v8. Thus, for crRNA complexes, we do not believe the two DNA nucleotides have significant impact and the results follow the expected logic of a single tracr molecule providing a more efficient platform.

- crHyb complexes show no statistical difference when complexes with Hyb v4 or Hyb v8. Hyb v4 was the template for tracrHyb v2 and therefore contains the same DNA nucleotides which would suggest the extra two DNA nucleotides of v8 would be unlikely to enhance activity. We would rather suspect the reduced activity on with tracrHyb.v2 is caused by extra constraints placed on this molecule by the more rigid backbone around stem loop 2, which further perturbs binding energy. It would be interesting to further probe this loop we feel this is beyond the scope of this manuscript. We have added a comment on this in the manuscript.

does the tracr3 half actually need to contain any RNA bases? I'm aware that some of those bases make contacts hydroxyl contacts with the protein, but did the authors ever try to fully remove all RNA from this portion of the tracr? It seems possible that this part of the tracr doesn't matter so much, especially given it can be truncated.

- While it is indeed true that the tracrRNA can be truncated in-vitro experiments, published work shows that for robust cellular activity requires a full length tracrRNA. However, this is a very good suggestion for the in-vitro exploration of backbone requirements. We have now tried a fully DNA 3'tracr region and it gives very similar activity to the hybrid version in the in-vitro assays. We have included this as a Supplementary Fig. 14 and include the following text in the results section:

"We inspected the Cas9:sgRNA crystal structure for potential steric clashes between the introduced thymine residues and the protein. All additional methyl groups were found over 5Å from the protein except the equivalent of nucleotide 90, where the group showed a clear clash (Supplementary Fig. 14a). However, it is possible that rotation of the backbone phosphate could induce a conformational change to relieve this clash, which correlate with the observed cleavage efficiencies. In fact, this region could be coded completely as DNA, with little effect on cleavage efficiencies (Supplementary Fig. 14b and c), again, suggesting potential clashes in this region are tolerated."

Minor comments

To make this work more accessible to a reader that is not intimately familiar with the structure of Cas9 and the relationship between the tracrRNA and crRNA, I would suggest adding a supplementary structure figure. If I read this paper pretending I don't know how these molecules interact, I find that the explanations are insufficient to really visualize the system and experiments. For example, when tracrRNA is first introduced it is not clear to an unfamiliar reader that the tracrRNA weaves through the Cas9 protein.

- We have added supplementary figure 1 to the introduction section depicting the overall structure of the complex.

Page 5, line 204: tracrRNA should be tracrRNA

- We have amended this mistake.

Page 5, lines 146-156: There is confusion in the writing here about whether residue 15 can be changed to DNA or not. At line 146 the text indicates that it can be changed, but then it is not tested in any of the combinations listed and is called a key residue on line 156. Looking at the figure, it does appear to have more of an effect than 1, 19, and 22, but the writing needs to be clarified in the main text for consistency.

- We thank the reviewer for this correction. The text lacked clarity and we have now amended to ensure accuracy and consistency. It now reads:

“In summary, we have identified three key residues within the crHyb molecule (residues 16, 23, 24) which cannot be coded as DNA. We have further described a fully active Hybrid design that contains only five RNA residues (residues 1, 15, 16, 23 and 24).”

Page 5, line 188 and Figure 3: It seems like the crR.Hyb contains RNA nucleotides in more than the RNA guide (target site recognition) region, as they also extend into the following RNA hairpin. Was a crR.Hyb tested that contained just the direct target site binding residues (and any others that are required, such as 23 and 24)? If not, what was the reasoning for extending the RNA residues past the target contact region? I don't think it's a necessary experiment, I just wonder about the reasoning or if there are additional data that weren't included in the manuscript.

- We have updated the text with a fuller explanation of the design decisions. We have not generated or tested a crHyb molecule which has the guide region fully coded as RNA. It would be interesting to test this design, but as the reviewers comment, we do not think it is critical data for the paper. The text now reads:

“To determine whether the reduced target binding affinity was caused by a reduction of RNA in guide or repeat sequence, we generated two additional crXNA molecules; crR.Hyb and crD.Hyb (Figure 3a). crR.Hyb coded only solvent accessible residues in the repeat-region as DNA. This assumed that there would be no conformational restraints imposed by the protein on this region and it could therefore tolerate the altered backbone properties. crD.Hyb consists of a predominantly DNA guide region and fully RNA repeat-region; see Figure 3a). Interestingly we found that target binding affinity appeared to be modulated by residues in the guide region.”

Page 7, line 279: I think this sentence needs a comma between “molecule” and “activity”, or needs to be rewritten.

- Many thanks for this correction.

Page 7, line 281: I would cite Figure 4 at the end of the sentence to remind readers where to find this data, and/or include numbers for comparison between the data in the two figures.

- We have updated the text as suggested.

Reviewer #3 (Remarks to the Author):

In this manuscript Rueda and coworkers investigated the importance of the ribonucleotides in the CRISPR molecule for Cas9 binding and activity. They produced several DNA-RNA hybrids of crRNA and tracrRNA containing deoxyribonucleotides instead of ribonucleotides at different positions of the molecules and tested their effect on Cas9 binding, DNA target recognition and Cas9 nuclease activity in vitro and in vivo. Based on these data, they identified critical regions of the Cas9 system that require ribonucleotides. Such work could be used for developing artificial CRISPR/Cas9 systems for making programmable endonuclease that could be used for therapeutic purpose. In that context, this work can be very valuable. However, a number of major points need to be addressed before this work is published.

Major points:

The authors performed several U to T substitutions and systematically concluded that the absence of 2'OH was responsible for the observed effects on Cas9 binding/activity. However, this substitution also introduces a methyl at position 5 of the base. The possible effect of this change should be evaluated using the structure of the Cas9:tracrRNA:crRNA complex bound to DNA and discussed.

- This is an excellent point and we thank the reviewer for this comment. We have carefully inspected the Cas9 crystal structure 4O08 to determine whether this methyl group can introduce any steric clashes with the protein. The majority of residues are Watson-Crick base paired ensuring a more fixed location within the structure, or in solvent exposed loops with no interaction with the protein. We have found only two potential sites. The modelled methyl group on uracil 80 would come within 4Å of the lysine749 amine group. This is not expected to cause issues as the lysine is not held in place and has freedom to move to accommodate this extra group. Movement of lysine749 is not expected to create any additional interactions or clashes. Modelling a methyl group onto Uracil 90 does show a clear clash with the sugar and base of nucleotide 89. A slight rotation along the axis of pyrimidine ring could avoid this clash, but this maybe unlikely due to the hydrogen bond with try981.
- To validate these findings we have performed additional in vitro cleavage assays using a crHyb where all thymine groups were replaced with deoxy Uracil groups. We find no difference in cleavage efficiencies between crHyb or the deoxy Uracil version, formally demonstrating the methyl group has no impact at these residues. We have added Supplementary Fig 4 with this data and the following text in the results section:

“The replacement of uracil with thymine bases introduces extra methyl groups into the crXNA molecule. This methyl group is not expected to impact on activity due to base stacking with target DNA or tracrRNA. The position of these groups within the crXNA was manually modelled within the Cas9 crystal structure² and we found that no methyl group would come within 5Å of protein, suggesting they would have no impact on the complex (data not shown). To validate these predictions we developed a qPCR assay to report on template depletion by Cas9 cleavage using guide sequences and a 1kb template sequence derived from the human AAVS1 locus. We generated a crHyb molecule and a crHyb.deoxy molecule where we replaced all thymine residues within crHyb with deoxy Uracil. This molecule would therefore not contain any additional methyl groups associated with thymine DNA bases. Assessing their ability to cleave target DNA, found template depletion over time or Cas9 protein concentration was unchanged when we used a Cas9:tracrRNA complexes associated with crHyb or a crHyb.deoxy (Supplementary Fig. 4). Thus, thymine within the crXNA does not appear to impact Cas9 cleavage activity.”

It is important to present raw data of cleavage activity assays (at least in supplemental data) in order to evaluate the quality of the results (e.g. Fig. 2B).

- We have included multiple supplementary figures with exemplar data of the cleavage assays (Supplementary figures 7, 9 and 12)

Dissociation curves of switchSENSE measurements are not shown (Fig. 3C and 5D). These results are important to judge about the quality of the data and visualize the absence of dissociation reported in the text.

- We have now included dissociation curves in Supplementary figures 10 and 14.

Given the binding variations observed in Fig. 3 between BLI and switchSENSE measurements, it is important to repeat the experiments at different concentrations of Cas9:tracrRNA:crRNA to obtain standard deviations.

- We have performed SwitchSENSE as well as BLI binding experiments in triplicates and now show the experimental errors.
- We have repeated the BLI measurements in Figure 3B and show the errors.
- We have also performed concentration-response binding experiments for cas9:tracrRNA:crRNA and were able to determine the pseudo-association rates in BLI experiments, shown in Supplementary Fig. 11.

In Fig. 5D the final plateau corresponding to the complete saturation of DNA molecules by Cas9:tracrRNA:crRNA should be visible to properly evaluate the fit of the data and the correct determination of the k_{on} .

- We have amended this figure and show the complete association curve in Figure 6D.

Minor point:

Figure 3C is not cited in the text.

- We thank the reviewer for careful reading. We have corrected this error.

REVIEWERS' COMMENTS:

Reviewer #1 (Remarks to the Author):

Besides numerous grammatical errors, my concerns regarding the manuscript have been addressed.

Reviewer #2 (Remarks to the Author):

The authors were thorough in addressing reviewer concerns. I have a couple of minor points remaining about the edits.

"Joung and colleagues have established a bacterial assay which could be adapted to allow evolution and rapid screening to identify Cas9 variants which function more efficiency with crXNA in vivo."

This particular bacterial selection system (as well as other in vivo directed evolution systems) depend on expressing the gRNA from a plasmid. It's not immediately clear to me how you could use this approach with crXNA. I would either add a brief explanation of how the method could realistically be modified to actually be used for this purpose, or suggest different approaches.

Minor comment: efficiency should be efficiently.

Pg 3, line 97: "To this aim, we here which could be exploited in future applications." There word "which" is used three times, and possess is used twice. I suggest rewriting this sentence and/or breaking it into two parts.

Figure 2 legend: What are the units on the 0.4, 0.2, 0.1? I may have missed it, but I could not locate it and it would be convenient to have in the legend. "Cas9 protein at 0.4 (solid line) 0.2 (long dash line) 0.1 (dot dash line) or 0.05 (dotted line)".

I might add a citation of the paper "Lessons from Enzyme Kinetics Reveal Specificity Principles for RNA-Guided Nucleases in RNA Interference and CRISPR-Based Genome Editing" mentioned in the reviewer response for the discussion of specificity, as it supports your findings and would be of interest to the people reading this work.

If Supplementary Fig. 2 causes a copyright issue, it should not be too challenging to generate something similar.

Reviewer #4 (Remarks to the Author):

Answers of the authors are convincing and appropriate experiments have been performed. I recommend the publication of this manuscript in Nature communication after the correction of these 2 minor points:

1. In their answer, the authors proposed to add the following text to the manuscript:
"The replacement of uracil with thymine bases introduces extra methyl groups into the crXNA molecule. This methyl group is not expected to impact on activity due to base stacking with target

DNA or tracrRNA. The position of these groups within the crXNA was manually modelled within the Cas9 crystal structure² and we found that no methyl group would come within 5Å of protein, suggesting they would have no impact on the complex (data not shown). To validate these predictions we developed a qPCR assay to report on template depletion by Cas9 cleavage using guide sequences and a 1kb template sequence derived from the human AAVS1 locus. We generated a crHyb molecule and a crHyb.deoxy molecule where we replaced all thymine residues within crHyb with deoxy Uracil. This molecule would therefore not contain any additional methyl groups associated with thymine DNA bases. Assessing their ability to cleave target DNA, found template depletion over time or Cas9 protein concentration was unchanged when we used a Cas9:tracrRNA complexes associated with crHyb or a crHyb.deoxy (Supplementary Fig. 4). Thus, thymine within the crXNA does not appear to impact Cas9 cleavage activity."

However, the following part of this text is not clear to me:

"This molecule would therefore not contain any additional methyl groups associated with thymine DNA bases. Assessing their ability to cleave target DNA, found template depletion over time or Cas9 protein concentration was unchanged when we used a Cas9:tracrRNA complexes associated with crHyb or a crHyb.deoxy (Supplementary Fig. 4)."

2. In sup. Figure 10: "time disassociation" on the x axis should be replaced by "time dissociation"

Please find below our responses to the reviewers comments.

REVIEWERS' COMMENTS:

Reviewer #1 (Remarks to the Author):

Besides numerous grammatical errors, my concerns regarding the manuscript have been addressed.

- All grammatical errors have now been corrected.

Reviewer #2 (Remarks to the Author):

The authors were thorough in addressing reviewer concerns. I have a couple of minor points remaining about the edits.

“Joung and colleagues have established a bacterial assay which could be adapted to allow evolution and rapid screening to identify Cas9 variants which function more efficiency with crXNA *in vivo*.”

This particular bacterial selection system (as well as other *in vivo* directed evolution systems) depend on expressing the gRNA from a plasmid. It's not immediately clear to me how you could use this approach with crXNA. I would either add a brief explanation of how the method could realistically be modified to actually be used for this purpose, or suggest different approaches.

- The Joung assay would require modifications to utilise synthetic gRNA rather than supplied by a plasmid. We have adjusted the text to the following:
“Joung and colleagues have established a bacterial assay²⁰ which could be adapted by electroporating synthetically generated crXNAs into bacterial cells inducibly expressing Cas9 variants libraries. This would allow rapid identification of Cas9 mutants that function more effectively with crXNA *in vivo*.”

Minor comment: efficiency should be efficiently.

- We have corrected this mistake.

Pg 3, line 97: “To this aim, we here . . . which could be exploited in future applications.” There word “which” is used three times, and possess is used twice. I suggest rewriting this sentence and/or breaking it into two parts.

- This was indeed poorly constructed. We have rewritten this sentence as follows.
- “To this aim, we here describe the first example of generation of hybrid DNA:RNA CRISPR and tracr molecules which direct specific Cas9 nuclease activity *in vitro* and

in vivo and possess unique biophysical properties that could be exploited in future applications.”

Figure 2 legend: What are the units on the 0.4, 0.2, 0.1? I may have missed it, but I could not locate it and it would be convenient to have in the legend. “Cas9 protein at 0.4 (solid line) 0.2 (long dash line) 0.1 (dot dash line) or 0.05 (dotted line)”.

- This was an oversight. The legend has now be corrected and units added.

I might add a citation of the paper “Lessons from Enzyme Kinetics Reveal Specificity Principles for RNA-Guided Nucleases in RNA Interference and CRISPR-Based Genome Editing” mentioned in the reviewer response for the discussion of specificity, as it supports your findings and would be of interest to the people reading this work.

- We have added this additional citation.

If Supplementary Fig. 2 causes a copyright issue, it should not be too challenging to generate something similar.

- We have obtained copy right permission for this figure.

Reviewer #4 (Remarks to the Author):

Answers of the authors are convincing and appropriate experiments have been performed. I recommend the publication of this manuscript in Nature communication after the correction of these 2 minor points:

1. In their answer, the authors proposed to add the following text to the manuscript: “The replacement of uracil with thymine bases introduces extra methyl groups into the crXNA molecule. This methyl group is not expected to impact on activity due to base stacking with target DNA or tracrRNA. The position of these groups within the crXNA was manually modelled within the Cas9 crystal structure² and we found that no methyl group would come within 5Å of protein, suggesting they would have no impact on the complex (data not shown). To validate these predictions we developed a qPCR assay to report on template depletion by Cas9 cleavage using guides sequences and a 1kb template sequence derived from the human AAVS1 locus. We generated a crHyb molecule and a crHyb.deoxy molecule where we replaced all thymine residues within crHyb with deoxy Uracil. This molecule would therefore not contain any additional methyl groups associated they thymine DNA bases. Assessing their ability to cleave target DNA, found template depletion over time or Cas9 protein concentration was unchanged when we used a Cas9:tracrRNA complexes associated with crHyb or a crHyb.deoxy (Supplementary Fig. 4). Thus, thymine within the crXNA does not appear to impact Cas9 cleavage activity.”

However, the following part of this text is not clear to me:

“This molecule would therefore not contain any additional methyl groups associated they thymine DNA bases. Assessing their ability to cleave target DNA, found template depletion over time or Cas9 protein concentration was unchanged when we used a Cas9:tracrRNA complexes associated with crHyb or a crHyb.deoxy (Supplementary Fig. 4).”

- We have adjusted the text for increased clarity. It now reads:
“We generated a crHyb.deoxy molecule where all thymine residues were replaced with deoxyUracil, to remove any additional methyl groups compared to crRNA. Assessing the ability of Cas9 complexes to cleave target DNA, we found template depletion over time was unchanged when Cas9:tracrRNA was complexed with crHyb or crHyb.deoxy, and when Cas9 protein concentration was varied (Supplementary Fig. 4). Thus, thymine within the crXNA does not appear to impact Cas9 cleavage activity.”

2. In sup. Figure 10: “time disassociation” on the x axis should be replaced by “time dissociation”

- We have updated the figure legend with this correction.